# $\mathbb{Z}_2$ topological orders in kagome dipolar systems: Feedback from Rydberg quantum simulator

Pengwei Zhao[1] and Gang v. Chen[1, 2, 3, *]

[1]*International Center for Quantum Materials, Peking University, Beijing 100871, China*
[2]*Tsung-Dao Lee Institute, Shanghai Jiao Tong University, Shanghai, 201210, China*
[3]*Collaborative Innovation Center of Quantum Matter, Beijing 100871, China*

(Dated: June 11, 2025)

The mutual feedback between quantum condensed matter and cold atom physics has been quite fruitful throughout history and continues to inspire ongoing research. Motivated by the recent activities on the quantum simulation of topological orders among the ultracold Rydberg atom arrays, we consider the possibility of searching for topological orders among the dipolar quantum magnets and polar molecules with a kagome lattice geometry. Together with other quantum interactions such as the transverse field, the dipolar interaction endows the kagome system with a similar structure as the Balents-Fisher-Girvin model and thus fosters the emergence of the $\mathbb{Z}_2$ topological orders. We construct a $\mathbb{Z}_2$ lattice gauge theory to access the topological ordered phase and describe the spinon and vison excitations for the $\mathbb{Z}_2$ topological orders. We explain the spectroscopic consequences for various quantum phases as well as the experimental detection. We further discuss the rare-earth kagome magnets, ultracold polar molecules, and cluster Mott insulators for the physical realization.

## I. INTRODUCTION

Ever since the proposal and discovery of the Bose-Einstein condensation, the mutual feedback and the confluence between the ultracold atomic systems and the quantum condensed matter physics have been a fertile ground for creativity and innovation [1–3]. In the new century of quantum simulation, this mutual interaction is further accelerated, generating a set of new ideas and activities [4–22]. One active direction of such activities is to search for the intrinsic topological order with the ultracold atom settings. Recent progress has been made towards the ultracold atom realization of the bosonic quantum Hall effect (chiral Abelian topological order) [14] and the $\mathbb{Z}_2$ topological order and the Rydberg atom arrays were designed for the latter [23, 24]. On the geometrically frustrated lattices, the Rydberg blockade creates the degenerate classical manifold. The detuning field generates the quantum tunneling between the classical states in the degenerate manifold and induces the topological orders with the assistance of the extended $1/r^6$ Rydberg interaction. Motivated by this interesting and important proposal, we consider the possibility of identifying the topological orders among the kagome dipolar systems.

The $\mathbb{Z}_2$ topological order was proposed for the Rydberg atom arrays on the kagome lattice [23, 24]. Owing to the Rydberg blockade and the long-range interaction, part of the physical intuition was drawn from making a connection to the well-known Balents-Fisher-Girvin (BFG) model on the kagome lattice [25]. The BFG model has a quantum dimer model (QDM) description on the triangular lattice formed by the centers of the hexagonal plaquettes [26, 27]. The equivalent dimer resonating is generated by the high-order perturbation by the detuning field

on the Rydberg atom. The numerical calculation found an extended region of $\mathbb{Z}_2$ spin liquid that is compatible with the physical intuition [23, 24]. The ingredients for making such a connection are the Rydberg blockade, the long-range interaction, and the geometrically frustrated kagome lattice. Along this line of thinking, it seems that the dipolar interactions on the frustrated lattice could carry these ingredients as well. Actually, the $1/r^3$ dipolar interaction is more extended than the $1/r^6$ interaction [21, 22]. It has been proposed recently a Dirac spin liquid can be realized in the kagome dipolar XY model [22]. We propose two different kagome dipolar systems. One is the dipolar quantum magnet with a kagome lattice geometry. This is naturally realized in the rare-earth-based tripod kagome compounds [28–30]. The other is the ultracold polar molecule on the kagome optical lattice [21, 31–36]. In fact, the ultracold polar molecule has been proposed to realize various spin models in quantum magnets as well as the extended $t$-$J$ models [31, 32]. Since both systems could share the same type of model Hamiltonian and the tripod kagome compounds involve more complications about the local moment degrees of freedom, we will devote most of our discussion to the kagome compounds, and postpone the discussion about the ultracold polar molecule and cluster Mott insulator realization at the end of this paper.

Differing from the $1/r^6$ Rydberg interaction, the dipolar interaction for the magnetic moments is not just more extended with a $1/r^3$ dependence, and moreover is anisotropic [21, 22]. We will begin with explaining the interaction and the physics by working with the local moments from the non-Kramers doublets for the tripod kagome systems, and then discuss the application to the Kramers doublets. For the rare-earth magnets, the strong spin-orbit coupling entangles the spin and the orbitals and leads to the spin-orbit-entangled moment $J$ [37]. Unlike the Kramers doublet whose degeneracy arises from the time-reversal symmetry for the half-

---
* chenxray@pku.edu.cn

integer $J$, the non-Kramers doublet occurs for the integer $J$ when the two-fold degenerate crystal field ground states are protected by the lattice symmetry. For the kagome lattice, however, no symmetries protect the degeneracy of the non-Kramers doublet. Hence, there always exists a finite splitting between the two states of the non-Kramers doublet on the kagome lattice [28]. If these two states of the non-Kramers doublet are well separated from other crystal field excited levels, the low-temperature magnetic physics is fully governed by the non-Kramers doublet. One then introduces an effective pseudospin-1/2 degree of freedom, $\boldsymbol{S}$, to operate on the non-Kramers doublet. An example of this kind is the $Ho^{3+}$ ion in the compound $Ho_3Mg_2Sb_3O_{14}$ [30]. Due to the microscopic reasons, only one component of the pseudospin, $S^z$, is linearly related to the magnetic dipole moment, and the splitting between the two states of the non-Kramers doublet can often be modeled as an intrinsic transverse field [28]. Thus, the relevant model for the system is given as

$$
\begin{aligned}
H = &\frac{1}{2} \sum_{i \neq j} \frac{V}{r_{ij}^3} \left[ \hat{\boldsymbol{z}}_i \cdot \hat{\boldsymbol{z}}_j - 3(\hat{\boldsymbol{z}}_i \cdot \hat{\boldsymbol{r}}_{ij})(\hat{\boldsymbol{z}}_j \cdot \hat{\boldsymbol{r}}_{ij}) \right] S_i^z S_j^z \\
&- \sum_i h_x S_i^x - \sum_i h_z (\hat{\boldsymbol{z}} \cdot \hat{\boldsymbol{z}}_i) S_i^z + \cdots ,
\end{aligned}
\tag{1}
$$

where the first term is the magnetic dipole-dipole interaction with $\boldsymbol{r}_{ij}$ the lattice vector connecting $i$ and $j$ and is long-ranged, the second term is the intrinsic transverse field and the third term is the external Zeeman field. The "$\cdots$" refers to the other interactions, especially the superexchange interaction between the pseudospins, and these interactions can sometimes help enhance quantum fluctuations. The magnetic moment of the $Ho^{3+}$ ion is quite large and is $\sim 9.74 \mu_B$ in $Ho_3Mg_2Sb_3O_{14}$ [30]. Hence, the dipole-dipole interaction is a large energy scale in the system and should be seriously considered. In Eq. (1), the $\hat{\boldsymbol{z}}_i$ is the local Ising direction at the lattice site $i$ and defines the direction of the local magnetization (see Fig. 1). By realizing the connection with the $\mathbb{Z}_2$ topological order and the BFG model, we adopt the $\mathbb{Z}_2$ parton-gauge construction for the $\mathbb{Z}_2$ topological ordered phases [23] (or $\mathbb{Z}_2$ spin liquids) and consider the instability towards other ordered phases. Throughout this work, we will use the $\mathbb{Z}_2$ topological orders and the $\mathbb{Z}_2$ spin liquids interchangeably. This parton-gauge construction not only captures the exotic properties within the $\mathbb{Z}_2$ topological order [23, 38–46], but also qualitatively establishes the global phase diagram.

While $\mathbb{Z}_2$ topological order has appeared in Rydberg atom experiments and quantum simulators, its evidence in real condensed matter materials has not yet been fully discovered in any existing experiment. Therefore, it is more important for us to address the key experimental features that distinguish it from other states. For this purpose, we discuss various experimental probes and their consequences. Besides the thermodynamic properties, we combine the microscopic properties of the local

moments and the symmetry enrichments of the $\mathbb{Z}_2$ topological order and explore the spectroscopics. Although there exist the spinon and the vison excitations for the $\mathbb{Z}_2$ topological order [47, 48], for the non-Kramers doublets, only the vison continuum is detectable in the inelastic neutron scattering measurements. This selective measurement arises from the microscopic property of the local moment and the relation of the magnetic moment with the vison bilinears [49, 50]. Moreover, the external Zeeman coupling polarizes the magnetic moment component $S^z$, and thus modulates the background dual $\mathbb{Z}_2$ gauge flux for the visons. In terms of the old language in the literature, distinct flux patterns correspond to odd and even $\mathbb{Z}_2$ gauge theories [47, 48, 51]. It is then shown that, at different magnetization plateaux, the system can access distinct symmetry enriched $\mathbb{Z}_2$ topological orders with distinct symmetry fractionalizations for the visons [43]. The direct consequence of the distinct symmetry fractionalization for the visons is the enhanced spectroscopic periodicity in the Brillouin zone.

Away from the context for the non-Kramers doublet, we further consider the Kramers doublet [52]. The intrinsic transverse field is not allowed by the time-reversal symmetry. The transverse field has to be applied externally. Once the system establishes the $\mathbb{Z}_2$ topological order, one can discuss the measurements. Unlike the non-Kramers doublet, both the spinon continuum and the vison continuum show up in the inelastic neutron scattering measurements. Due to the clear separation of the spinons and the visons in energies, it is feasible to separately identify their contribution and examine the spectroscopic signatures. In the ultracold polar molecule context, however, the measurement of the excitations is not as straightforward as the inelastic neutron scattering measurement in quantum materials [53, 54]. There is the spin-echo type of measurement like NMR for the polar molecules and the two-photon Raman spectroscopy that is equivalent to the inelastic neutron scattering measurement [31, 32]. Unlike the quantum materials whose couplings are more-or-less fixed by the materials themselves, the advantage of the polar molecule quantum simulation is to provide more tunability of the couplings in the model Hamiltonian.

For the realization with the cluster Mott insulators, the electron charge is identified as the relevant degree of freedom [55–58]. The electron's occupation and absence on a lattice site can be thought of as an effective spin-1/2 degree of freedom. The zero-field case of the spin problem corresponds to the 1/4 filling of the electrons. Unlike the spin system, here the electron spectral function encodes the signature of charge fractionalization, and density-density correlation encodes the vison continuum.

The remaining parts of the paper are organized as follows. In Sec. II, we introduce the model formulation for both non-Kramers doublet and Kramers doublet on the kagome lattice, as well as the ultracold polar molecule context. In Sec. III, we briefly explain the relation be-

tween our model and the BFG model and compare the difference from the Rydberg atom array. In Sec. IV, we perform a perturbative analysis of our model and gain some basic insights about the condition for $\mathbb{Z}_2$ topological order. In Sec. V, we work out the global phase diagram by a non-perturbative parton-gauge mean-field theory. In Sec. VI, we devote to the physical properties of the $\mathbb{Z}_2$ topological ordered phases, and these include the detailed excitation structures for both the spinon continuum and the vison continuum. Finally in Sec. VII, we elaborate on the detection of the $\mathbb{Z}_2$ topological orders from both thermodynamic and spectroscopic measurements and discuss the real materials (including the rare-earth magnets and the cluster Mott insulators) and the feasibility of quantum simulation with the ultracold polar molecules.

## II. MODEL FORMULATION

We start with the non-Kramers doublet on the kagome dipolar magnets. In the actual tripod kagome material, the local easy axis is controlled by the local crystal environment and the non-magnetic ions above/below the kagome plane, and thus can be tuned by ambient pressure or chemical pressure. To avoid the complication with the structural details, we work in the case that each local spin moment is aligned perpendicular to the kagome lattice plane with a uniform $\hat{z}$ on every site. The variation from a uniform $\hat{z}$ is discussed at the end of the paper. Under this choice, the model in Eq. (1) is simplified to

$$H = \frac{1}{2}\sum_{i \neq j} V_{ij} S_i^z S_j^z - h_z \sum_i S_i^z - h_x \sum_i S_i^x + \cdots , \quad (2)$$

where $V_{ij} = g/r_{ij}^3 + J_{ij}^z$ has included both the dipolar interaction and the superexchange interaction $J_{ij}$, and "$\cdots$" refers to other interactions such as the superexchange between the transverse spin components, $J_{ij}^{\perp}(S_i^x S_j^x + S_i^y S_j^y)$. Here, $g$ refers to the coupling strength of the dipolar interaction. Since we prefer the local moment to be in the Ising limit (with the doublet wavefunction to be polarized towards the large $J$ states), the superexchange between the transverse spin components could be relatively weak.

As we have previously mentioned, the degeneracy of non-Kramers doublet is not protected by the symmetry of the kagome lattice and the time reversal, allowing for the introduction of an intrinsic transverse field, $h_x$, to account for the intrinsic splitting of the non-Kramers doublets. Under the time-reversal operator, the effective pseudospin-1/2 operator for the non-Kramers doublet transforms as

$$\mathcal{T}(S^x, S^y, S^z) \overset{\text{non-Kramers}}{\Longrightarrow} (S^x, S^y, -S^z). \quad (3)$$

Thus, only $S^z$ of the non-Kramers doublets couples to the external Zeeman field, $h_z$. As we will clarify later, this unique property of non-Kramers doublets leads to a selective measurement of the vison excitations for the

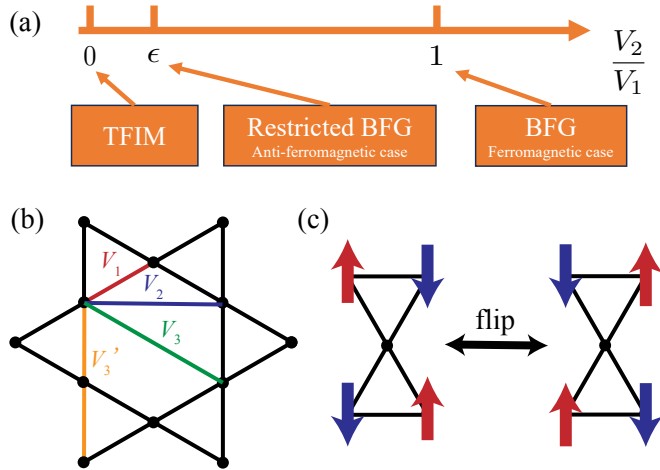

FIG. 1. (a) A schematic phase diagram for kagome dipolar magnets is presented. We assume $V_2 = V_3$ in this plot. When $V_2$ and $V_3$ are fully suppressed, the model reduces to a transverse-field Ising model (TFIM) on the kagome lattice. However, if $V_2$ and $V_3$ acquire finite values relative to $V_1$, the system flows to a restricted BFG regime, which can be mapped to the QDM with extra constraints. When the strengths of $V_2$ and $V_3$ become comparable to $V_1$, the model flows to the conventional BFG regime. (b) Definition of interactions between the $n$th neighbor sites. (c) Spin representation of the flippable dimer structure and dimer flipping.

$\mathbb{Z}_2$ topological ordered states in the measurement such as the inelastic neutron scattering (INS) experiments.

For the Kramers doublet, the two-fold degeneracy is protected by the time-reversal symmetry. In this case, the transverse field $h_x$ has to be applied externally. The effective pseudospin-1/2 operator transforms like a real magnetic moment under the time-reversal operator,

$$\mathcal{T}(S^x, S^y, S^z) \overset{\text{Kramers}}{\Longrightarrow} (-S^x, -S^y, -S^z). \quad (4)$$

As a result, the INS-like experiments can probe more types of fractionalized excitations in the $\mathbb{Z}_2$ topological ordered states.

The extended dipolar interaction can be further realized with the ultracold polar molecules on optical lattices. A polar molecule consists of two atoms that are often alkali-metal atoms but different atoms. As the diatomic object breaks the spatial rotational symmetry, one needs the rotational degrees of freedom to describe a polar molecule. The rotational degrees of freedom endow the polar molecules with permanent dipole moments, whose direction can be controlled by an external electric field. When they are loaded onto an optical kagome lattice, these polar molecules interact by a dipolar interaction. Such a setup leads to a $t$-$J$-$V$-$W$ model [31, 32], whose coupling coefficients can be tuned experimentally. Under the suitable circumstances, the Hamiltonian in Eq. (2) can be reproduced.

## III. CONNECTION TO THE BALENTS-FISHER-GIRVIN MODEL

What is the relation between our model and the BFG model? We here sketch a description of the BFG model [25]. In the theoretical progress of spin liquids, one goal was to go beyond the contrived models and write down the physical-looking models such that these models may be experimentally relevant and then spin liquids can be realized physically. The BFG model serves as one of the first few such physical-looking models around its time. It incorporates first-neighbor, second-neighbor, and third-neighbor Ising interactions, where the third-neighbor is between the diagonal sites on the hexagon plaquette of the kagome lattice [see Fig. 1(b)]. When these three interactions are equal, the Ising part of the interaction can be expressed in terms of the square of the total $S^z$ spin within each hexagon, leading to a highly degenerate ground state manifold. With the perturbative exchange interaction between the transverse spin components, the system develops a 4-site ring exchange [see Fig. 1(c)] interaction that allows quantum fluctuations among the degenerate manifold, and these quantum processes can be mapped to the dimer resonances for a QDM on the triangular lattice formed by the centers of the hexagon plaquettes. It is well established that the QDM on a triangular lattice exhibits a $\mathbb{Z}_2$ topological ordered phase near the Rokhsar-Kivelson point [26, 42, 59], and thus, the BFG is one of the first few physical-looking models with a $\mathbb{Z}_2$ topological ordered ground state.

To clarify the relation of our model with the BFG model, we first let $V_n$ represent the Ising interaction strength $V_{ij}$ between the $n$th neighbor sites. With only the nearest-neighbor Ising interaction $V_1$, the model reduces to the kagome transverse field Ising model (TFIM). The antiferromagnetic kagome TFIM at zero longitudinal field $h_z$ is known to be a quantum paramagnet and is smoothly connected to the paramagnetic phase at the strong transverse fields [25]. There is no topological order in the absence of long-range interactions.

In the superexchange interactions $J_{ij}^z$'s, the ratio of the long-range dipolar interactions is given by $V_1 : V_2 : V_3 : V_4 = 1 : 0.193 : 0.125 : 0.054$. Here, $V_4$ is comparably weak and may be truncated in the first analysis. It is illuminating to compare this ratio with the one for the Rydberg atom array that is $1 : 1/27 : 1/64$. Although the ratio for the Rydberg atoms is far away from the uniform limit in the original BFG model, the ring exchange in the BFG model can be generated by the fourth-order perturbation of the transverse field term, and the non-uniformity of the Ising interaction does not play a significant role in destabilizing the $\mathbb{Z}_2$ topological order for the Rydberg atom array [24]. In contrast, our kagome dipolar model is actually less non-uniform than the Rydberg atom array and is probably more favorable toward the BFG model realization and the associated $\mathbb{Z}_2$ topological order.

We summarize and extend our qualitative discussion above in the schematic plot of Fig. 1(a). This plot is not a phase diagram and should not be taken quantitatively. It can only be used for the qualitative understanding. For the convenience of the explanation, we set $V_2 = V_3$ and the other longer-distance Ising interactions to zero. In the limit of $V_2/V_1 \to 0$, the system is in the TFIM regime and the quantum paramagnetic state. With a weak and finite $V_2/V_1$, the system enters the so-called "restricted BFG regime" where the large $V_1$ Ising interaction *restricts* the spin configuration on each triangular plaquette to be ↑↑↓ or ↓↓↑ and the remaining interactions still drive the system with the restricted states into the topological phases of the BFG model. The Rydberg atom array on the kagome lattice is suggested to be related to this "restricted BFG regime". When $V_2/V_1 = 1$, the system can be well mapped to the BFG model. This regime is referred to as the "BFG regime" in the plot. With the dipole-dipole interaction, the kagome dipolar magnet, owing to the presence of the nearest-neighbor superexchange interaction, can be driven to the "restricted BFG regime" or "the BFG regime". This depends on the sign of the nearest-neighbor Ising exchange coupling, which we explain below.

## IV. PERTURBATIVE ANALYSIS FOR $\mathbb{Z}_2$ TOPOLOGICAL ORDER

In this section, we provide a perturbative analysis of the model in Eq. (2). Although the validity of the perturbative analysis is limited to the perturbation treatment, it gives us a basic understanding of the emergence and the possible conditions for the $\mathbb{Z}_2$ topological order. We separate the discussion into two subsections. The first subsection is about the ferromagnetic Ising exchange case and the second subsection is about the antiferromagnetic Ising exchange case.

### A. Ferromagnetic case

Unlike the dipolar interaction that is extended and long-ranged, the superexchange interaction of the $4f$ electrons is quite short-ranged and is often dominated by the first exchange [60, 61]. Here, we consider the effect for a ferromagnetic Ising exchange with $J_1^z < 0$ between the first neighbors. In this case, the total first-neighbor Ising interaction, $V_1$, is suppressed, and the ratio to second and third-neighbor Ising interactions is more toward uniformity. Thus, the model of Eq. (2) in this case is closer to the original BFG model than the Rydberg atom array.

Ignoring the longer range interactions for $(V_{n \geq 4})$, the Hamiltonian in Eq. (2) can be written as

$$H = H_{\bigcirc}^{\mathrm{f}} + H_{\sim}^{\mathrm{f}} + H_{3'} + H_x + \cdots , \qquad (5)$$

where

$$H_\bigcirc^{\mathrm{f}} = \frac{V_3}{2} \sum_{\bigcirc_{\mathbf{r}}} \left( S_{\mathbf{r}}^z - \frac{h_z}{2V_3} \right)^2, \tag{6}$$

$$H_\sim^{\mathrm{f}} = (V_1 - V_3) \sum_{\langle ij \rangle} S_i^z S_j^z + (V_2 - V_3) \sum_{\langle\langle ij \rangle\rangle} S_i^z S_j^z, \tag{7}$$

$$H_{3'} = V_3 \sum_{\langle\langle\langle ij \rangle\rangle\rangle'} S_i^z S_j^z, \tag{8}$$

$$H_x = -h_x \sum_i S_i^x. \tag{9}$$

Here, $S_{\mathbf{r}}^z = \sum_{i \in \bigcirc_{\mathbf{r}}} S_i^z$ represents the total $S^z$ of the spins on the six corners of the hexagon plaquette, and $\bigcirc_{\mathbf{r}}$ refers to the hexagon that is centered at $\mathbf{r}$. The term $H_\bigcirc^{\mathrm{f}}$ corresponds to the BFG Hamiltonian in the presence of an out-of-plane magnetic field, $H_\sim^{\mathrm{f}}$ captures all spatial nonuniformity in the interactions, $H_{3'}$ accounts for the inter-hexagon third-neighbor Ising interaction, $H_x$ represents the transverse field, and "$\cdots$" refers to the remaining interaction such as the superexchange between the transverse components and will not be included in our calculation of the main text.

If $H_\sim^{\mathrm{f}} = H_{3'} = 0$ is satisfied, the system reduces to the BFG model. Depending on $h_z$, several magnetization plateaux can emerge in $H_\bigcirc^{\mathrm{f}}$ with the magnetization $m_z = 0, 1/6, 1/3$ of the saturation magnetization $m_{\mathrm{sat}}$. The $m_z = 0$ plateau corresponds to the case that is studied in the original BFG model [25]. Up to the leading-order perturbation in $H_x$, the effective Hamiltonian becomes a 3-dimer QDM with the dimers connecting the centers of the hexagon plaquette [24]. Here, the hexagon centers form a triangular lattice, and the dimer covering (absence) on the bonds of the triangular lattice corresponds to the spin-$\downarrow$ ($\uparrow$) configuration on the kagome lattice. The $m_z = 0$ plateau leads to the three dimers connecting every triangular site. Similarly, for $m_z = m_{\mathrm{sat}}/6$ and $m_z = m_{\mathrm{sat}}/3$, the effective Hamiltonian reduces to the 2-dimer and 1-dimer QDMs on the triangular lattice, respectively [24, 43].

In all three cases, the effective Hamiltonian takes the following form

$$\begin{aligned} H_{\mathrm{eff}} = &-t \sum_{\boxslash} \left( |\boxslash\rangle \langle\boxbslash| + |\boxbslash\rangle \langle\boxslash| \right) \\ &+ \mu \sum_{\boxslash} \left( |\boxslash\rangle \langle\boxslash| + |\boxbslash\rangle \langle\boxbslash| \right), \end{aligned} \tag{10}$$

where the only difference between the cases lies in the dimer constraint. The dimer-resonating strength is given by $t \sim \mathcal{O}(h_x^4/V_3^3)$, while $\mu$ represents the on-site potential for the flippable dimer structures, which is $\mu = 0$ in the cases here. Moreover, the neglected transverse exchange $J_1^\perp (S_i^x S_j^x + S_i^y S_j^y)$ contributes to the dimer resonance at the second order, and this further enhances the energy scale of the dimer-resonating and may be regarded as one advantage over the Rydberg atom array

[23, 24]. The triangular lattice QDM is known to host an extended $\mathbb{Z}_2$ topological ordered phase near the Rokhsar-Kivelson point [26]. Quantum Monte Carlo simulations in Ref. [43] suggest that for $m_z = 0$, the liquid phase remains extended even for $\mu = 0$. For $m_z = m_{\mathrm{sat}}/6$, a finite on-site potential $\mu$ is needed to stabilize the $\mathbb{Z}_2$ topological order [43]. For $m_z = m_{\mathrm{sat}}/3$, recent numerical simulations suggest that the $\mathbb{Z}_2$ liquid phase also lives in the finite $\mu$ region [62].

In the previous discussion, we have neglected the deformation term $H_\sim^{\mathrm{f}}$ and the inter-hexagon third-nearest-neighbor interaction $H_{3'}$. The deformation can be safely ignored if both $V_1 - V_3 \ll V_3$ and $V_2 - V_3 \ll V_3$ [24]. Nevertheless, $H_{3'}$ plays a critical role in stabilizing the liquid phase. Since the strength of $H_{3'}$ can be comparable to $V_3$, the ground state manifold is significantly altered. A sufficiently strong $H_{3'}$ suppresses the presence of the flippable dimer structures, thus destroying the topological order and triggering a phase transition into the Ising ordered phases. Therefore, stabilizing the $\mathbb{Z}_2$ topological order prefers weakening the strength of $H_{3'}$.

At the level of the perturbative analysis, we have already noticed that a weak $H_{3'}$ helps stabilize the topological order. One potential solution is to involve the superexchange interactions. On the kagome lattice, due to the different exchange paths, the inter-hexagon and intra-hexagon third-neighbor superexchange interactions should differ from each other. The other solution is to consider the noncolinear local axis of the dipole moments. As we will discuss in the end, the tilting of the local axis away from $\hat{\boldsymbol{z}}$ could significantly suppress $H_{3'}$. These together could potentially weaken $H_{3'}$, expressed as

$$H_{3'} = V_3' \sum_{\langle\langle\langle ij \rangle\rangle\rangle'} S_i^z S_j^z, \tag{11}$$

where $V_3'$ incorporate other effects such that $V_3'$ could be much weaker than $V_3$. Under these conditions, the effective Hamiltonian from the perturbative analysis derived earlier remains valid. Moreover, for $m_z = m_{\mathrm{sat}}/6$, the weak $V_3'$ interaction behaves as an on-site potential $\mu = V_3'$, which nevertheless could help stabilize the topological order.

## B. Antiferromagnetic case

For the second case, $J_1^z$ is antiferromagnetic, and $V_1$ becomes much stronger than the higher-order interactions, such that $V_1 \gg V_2 \gtrsim V_3 \gg V_{n\geq 4}$. The strong nearest-neighbor interaction $V_1$ imposes a restriction on the spin configurations of the ground states. If the external field $h_z$ is relatively small compared to $V_1$, the spin configuration on each triangular plaquette must be $\uparrow\uparrow\downarrow$ or $\downarrow\downarrow\uparrow$. In such a restricted space, the remaining interactions still lead to the BFG model. Thus, we refer to this case as the "restricted BFG regime" (see Fig. 1).

To gain more understanding of the restricted BFG

model, we rewrite the model in Eq. (2) as

$$H = H_\triangle^{\rm af} + H_\bigcirc^{\rm af} + H_\sim^{\rm af} + H_{3'} + H_x + \cdots, \qquad (12)$$

where

$$H_\triangle^{\rm af} = \frac{V_1 - V_3}{2} \sum_{\triangle_{\bm R}} \left[ S_{\bm R}^z - \frac{h_z}{2(V_1 - V_3)} \right]^2, \qquad (13)$$

$$H_\bigcirc^{\rm af} = \frac{V_3}{2} \sum_{\bigcirc_{\bm r}} (S_{\bm r}^z)^2, \qquad (14)$$

$$H_\sim^{\rm af} = (V_2 - V_3) \sum_{\langle\langle ij \rangle\rangle} S_i^z S_j^z. \qquad (15)$$

The terms $H_{3'}, H_x, \cdots$ remain the same as in the ferromagnetic case, and $H_{3'}$ also includes the superexchange interaction. Here, $S_{\bm R}^z = \sum_{\triangle_{\bm R}} S_i^z$ is the total $S^z$ of the spins on the three corners of the triangular plaquette $\triangle_{\bm R}$, which is centered as $\bm R$. These $\bm R$'s form a honeycomb lattice, which is the dual lattice of the triangular lattice formed by these $\bm r$'s (see Fig. 2).

For sufficiently strong antiferromagnetic $J_1^z$, the energy scale of $V_1$ becomes much larger than the other interactions in the system. Since $V_1 - V_3 \gg V_3$ and $H_\triangle^{\rm af}$ is the largest energy scales of the model, we restrict the discussion to its ground states manifold. The external magnetic field $h_z$ modifies the magnetization of the ground states, leading to the magnetic plateaux $m_z = \pm 1/3, \pm 1$ of the saturation magnetization $m_{\rm sat}$. If $h_z$ lies deep in the magnetic plateaux, the number of down-spins in each triangle is fixed. For example, in the $1/3$ magnetization, each triangle is $\uparrow\uparrow\downarrow$ state and the number of down-spins (the number of dimers) is one. Then, we consider $H_\bigcirc^{\rm af}$, which is exactly an BFG Hamiltonian. However, because a dimer-flipping term (the 1st row of Eq. (10)) now changes the number of down-spins in the triangles, the QDM description cannot be realized in these cases. We have to consider those cases in which the number of down-spins is allowed to vary. These correspond to that $h_z$ is near the transition between two magnetization plateaux.

Concretely, we consider the case $h_z \approx 0$, in which the ground states manifold consists of those $\uparrow\uparrow\downarrow$ and $\downarrow\downarrow\uparrow$ configurations, which can be understood as a restriction of the Hilbert space. Further considering the remaining terms: $H_\bigcirc^{\rm af}$, $H_\sim^{\rm aff}$, $H_{3'}$, more detailed structures would appear in the restricted Hilbert space. One can notice that $H_\bigcirc^{\rm af}$ takes the same form as the BFG model. After ignoring the deformation $H_\sim^{\rm aff}$ and $H_{3'}$ and regarding the down-spins as dimer coverings, the quantum dynamics of $H_x$ on the ground states of $H_\bigcirc^{\rm af}$ is captured by the same QDM as Eq. (10), while a constraint should be imposed: the number of dimers on each triangle is either 1 or 2.

The QDM on a triangular lattice hosts an extended $\mathbb{Z}_2$ liquid phase, and a chemical potential term can be added to further stabilize this phase. The key question here is whether the extra constraint imposed on the dimers

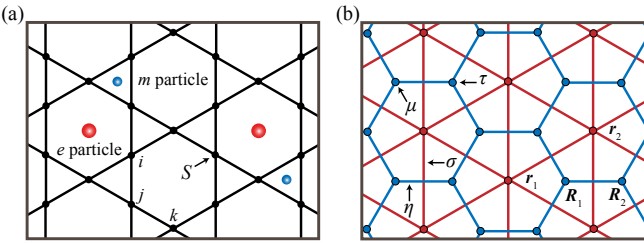

FIG. 2. (a) The kagome lattice sites are denoted as $i, j, k, \ldots$. The $\mathbb{Z}_2$ QSL of a kagome dipolar magnet contains two elementary excitations: $e$ particles (spinons) and $m$ particles (visons). $e$ particles (in red) live inside the hexagonal plaquettes; $m$ particles (in blue) live inside the triangular plaquettes. (b) A triangular lattice (in red) is constructed by connecting the hexagon centers of the kagome lattice, and its lattice sites are denoted as $\bm r_1, \bm r_2, \ldots$. The $\tau$ matter field and the $\sigma$ gauge field are placed on the triangular lattice sites and links, respectively. The dual lattice of the triangular lattice is the honeycomb lattice (in blue), where a dual $\mu$ matter field and a dual $\eta$ gauge field are put on the sites and links, respectively. The honeycomb lattice sites are denoted as $\bm R_1, \bm R_2, \ldots$.

prevents the formation of a liquid phase in the QDM. The answer can be argued by examining the Rokhsar-Kivelson (RK) point (with $\mu = t$),

$$H_{\rm RK} = -t \sum_{\triangledown} \left( |\square\!\!\!\square\rangle - |\square\!\!\!\square\rangle \right) \left( \langle\square\!\!\!\square| - \langle\square\!\!\!\square| \right), \qquad (16)$$

where the QDM is exactly solvable. The solution is the equal-amplitude superposition of all dimer coverings with the constraint. In such a state, one can break one dimer and create two spinons (or monomers). The energy of the state is independent of the separation between two spinons. Therefore, the spinons are deconfined. The constraint does not destroy the $\mathbb{Z}_2$ liquid phase of the QDM.

## V. NON-PERTURBATIVE MEAN-FIELD THEORY

### A. Parton-gauge construction

Beyond the perturbation theory, we study the $\mathbb{Z}_2$ spin liquid in our model by the non-perturbative mean field theory. The framework has been developed and well-tested in the Rydberg atom array [23]. The formulation begins with the parton-gauge construction for the $\mathbb{Z}_2$ spin liquid. Ignoring the gapped particles in the $\mathbb{Z}_2$ topological order, the remaining are the gauge links [23]. We first introduce the gauge links and then insert the gapped particles. In the spirit of the string-net condensation [63], the loops of the gauge links are closed strings that are condensed, and the ends of the open strings are deconfined particles due to the tensionless strings. The spin operator $\bm S$ corresponds to the shortest open string. In

terms of the gauge links, the spin operator $\boldsymbol{S}$ is set to

$$S_i^x \to \frac{1}{2}\sigma_{\boldsymbol{r_1 r_2}}^z, \tag{17}$$

$$S_i^z \to \frac{1}{2}\sigma_{\boldsymbol{r_1 r_2}}^x, \tag{18}$$

where $\boldsymbol{r_1 r_2}$ is the link on the triangular lattice that is formed by the hexagon centers of the kagome lattice, and the $i$ is the kagome lattice site that is located at the midpoint of $\boldsymbol{r_1 r_2}$ (see Fig. 2). Here, $\sigma^x$ and $\sigma^z$ are Pauli matrices. A $\tau$ field is then introduced as the spinon matter field at the hexagon centers of the kagome lattice and attached to the end of the open string. The original spin model is then reformulated as a $\mathbb{Z}_2$ lattice gauge theory on the triangular lattice [23] with

$$\begin{aligned}
H = \frac{1}{2}\sum_{\boldsymbol{r_1 r_2}, \boldsymbol{r_3 r_4}}^{\boldsymbol{r_1 r_2} \neq \boldsymbol{r_3 r_4}} \frac{V_{\boldsymbol{r_1 r_2}, \boldsymbol{r_3 r_4}}}{4}\sigma_{\boldsymbol{r_1 r_2}}^x \sigma_{\boldsymbol{r_3 r_4}}^x \\
+ \frac{h_z}{2}\sum_{\boldsymbol{r_1 r_2}}\sigma_{\boldsymbol{r_1 r_2}}^x + \frac{h_x}{2}\sum_{\boldsymbol{r_1 r_2}}\sigma_{\boldsymbol{r_1 r_2}}^z \tau_{\boldsymbol{r_1}}^z \tau_{\boldsymbol{r_2}}^z.
\end{aligned} \tag{19}$$

To constrain the enlarged Hilbert space, we impose the Gauss's law

$$\tau_{\boldsymbol{r}}^x = \prod_{\boldsymbol{rr'} \in \bigstar_{\boldsymbol{r}}} \sigma_{\boldsymbol{rr'}}^x, \tag{20}$$

where we have denoted the triangular lattice sites as $\boldsymbol{r}$, and $\bigstar_{\boldsymbol{r}}$ is the set of six legs emanating from $\boldsymbol{r}$. The notation $V_{\boldsymbol{r_1 r_2}, \boldsymbol{r_3 r_4}}$ is the same as the original $V_{ij}$, where $i$ or $j$ is the center of the link $\boldsymbol{r_1 r_2}$ or $\boldsymbol{r_3 r_4}$.

We here give more explanation of the physical meaning for the $\tau$ field and the $\sigma$ field [23]. From Eq. (20), $\tau_{\boldsymbol{r}}^x$ records the $\mathbb{Z}_2$ gauge charge that creates the gauge field around $\boldsymbol{r}$ as the Gauss's law. Given a ground state with a determined gauge charge distribution, the action of $S_i^x$ on the kagome lattice site $i$ changes the sign of gauge field $\sigma_{\boldsymbol{rr'}}^x = 2S_i^z$. Thus, the $\mathbb{Z}_2$ gauge charge $\tau_{\boldsymbol{r}}^x$ and $\tau_{\boldsymbol{r'}}^x$ are flipped, creating two point-like excitations. Continuous action of $S_i^x$ along a path in the kagome lattice can move the two excitations away from each other, and the two ends of this path/string are two gapped excitations. Thus, the $\tau$ field is interpreted as the spinon field. A similar interpretation could be made for the visons via the $S^z$ strings. $\tau_{\boldsymbol{r}}^x$ records the $\mathbb{Z}_2$ spinon parity and $\tau_{\boldsymbol{r}}^z$ is the spinon operator. Spinons are minimally coupled to the gauge link $\sigma^z$-field. When the gauge field is in the deconfined state with a nonzero $\langle \sigma^z \rangle$, separating two spinons creates a string that has no tension, and the spinons are deconfined in this case [23, 64]. When the spinon becomes gapless and condensed, the system is in a Higgs phase. Besides the deconfined and Higgs phases, there is the confining phase that is realized when the vison is condensed [23, 64].

## B. Mean-field theory

To develop a mean-field theory and access the $\mathbb{Z}_2$ spin liquid, we introduce a hardcore boson representation of the spin-1/2 operators [23, 43]. The gauge field is expressed as

$$\sigma_{\boldsymbol{rr'}}^x = 1 - 2B_{\boldsymbol{rr'}}^\dagger B_{\boldsymbol{rr'}}, \tag{21}$$

$$\sigma_{\boldsymbol{rr'}}^z = B_{\boldsymbol{rr'}}^\dagger + B_{\boldsymbol{rr'}}, \tag{22}$$

where $B_{\langle ij \rangle}$ and $B_{\boldsymbol{rr'}}^\dagger$ are hardcore boson operators. In this representation, spin-up (down) states are mapped to zero (one) boson states. For the spinon $\tau$ field, we choose a slightly different substitution. We aim for the ground state to be free of gauge excitations. For instance, if the ground state enforces $\tau_{\boldsymbol{r}}^z = 1$ ($\tau_{\boldsymbol{r}}^z = -1$), then $\tau_{\boldsymbol{r}}^z = -1$ ($\tau_{\boldsymbol{r}}^z = 1$) will represent an excitation. Thus, we define

$$\tau_{\boldsymbol{r}}^x = \zeta_{\boldsymbol{r}}(1 - 2b_{\boldsymbol{r}}^\dagger b_{\boldsymbol{r}}), \tag{23}$$

$$\tau_{\boldsymbol{r}}^z = b_{\boldsymbol{r}}^\dagger + b_{\boldsymbol{r}}, \tag{24}$$

where $\zeta_{\boldsymbol{r}} = 1$ or $\zeta_{\boldsymbol{r}} = -1$. Assuming the translational symmetry is protected, we focus on two types of ground states: (1) an even $\mathbb{Z}_2$ spin liquid, where $\tau_{\boldsymbol{r}}^z = 1$ for all $\boldsymbol{r}$, and we set $\zeta_{\boldsymbol{r}} = 1$; and (2) an odd $\mathbb{Z}_2$ spin liquid, where $\tau_{\boldsymbol{r}}^z = -1$ for all hexagons, and we set $\zeta_{\boldsymbol{r}} = -1$. Consequently, we can omit the $\boldsymbol{r}$-dependence of $\zeta_{\boldsymbol{r}}$.

In the hardcore boson representation, the $\mathbb{Z}_2$ gauge charge is represented by the number of bosons. The gauge constraint of Eq. (20) can be written as

$$b_{\boldsymbol{r}}^\dagger b_{\boldsymbol{r}} + \sum_{\boldsymbol{rr'} \in \bigstar_{\boldsymbol{r}}} B_{\boldsymbol{rr'}}^\dagger B_{\boldsymbol{rr'}} = Q, \tag{25}$$

$$\text{with} \quad Q = 0, 1, 2, \ldots, 7. \tag{26}$$

For an even (odd) $\mathbb{Z}_2$ spin liquid, $Q$ is even (odd). This gauge constraint is enforced by a Lagrange multiplier $\lambda$, leading to an interacting theory between the $B$ and $b$ bosons. To determine the dispersion of the $b$ bosons, we condense the $B$ bosons, setting

$$\langle B_{\boldsymbol{rr'}} \rangle = \langle B_{\boldsymbol{rr'}}^\dagger \rangle = \mathcal{B}. \tag{27}$$

This results in a Bogoliubov-de-Gennes (BdG) Hamiltonian for the spinons

$$\begin{aligned}
\mathcal{H}_{\text{spinon}} = -h_x \mathcal{B}\sum_{\boldsymbol{rr'}}(b_{\boldsymbol{r}}^\dagger + b_{\boldsymbol{r}})(b_{\boldsymbol{r'}}^\dagger + b_{\boldsymbol{r'}}) \\
+ \lambda \sum_{\boldsymbol{r}} b_{\boldsymbol{r}}^\dagger b_{\boldsymbol{r}} + E_0,
\end{aligned} \tag{28}$$

where

$$\begin{aligned}
E_0 = \frac{\mathcal{V}}{4}N_{\text{k}}(1 - 2\mathcal{B}^2)^2 - \frac{h_z}{2}N_{\text{k}}(1 - 2\mathcal{B}^2) \\
+ \lambda N_{\text{t}}(6\mathcal{B}^2 - Q),
\end{aligned} \tag{29}$$

where $\mathcal{V} = 2V_1 + 2V_2 + 3V_3 + \ldots$ represents the total dipolar interaction energy per spin, and $N_{\text{k}} = 3N_{\text{t}}$,

with $N_k$ and $N_t$ denoting the number of sites on the kagome and triangular lattices, respectively. Diagonalizing $\mathcal{H}_{\mathrm{spinon}}$ in momentum space as [65–68]

$$\mathcal{H}_{\mathrm{spinon}} = \sum_{\boldsymbol{k}}^{\mathrm{B.Z.}} \omega_{\boldsymbol{k}}(a_{\boldsymbol{k}}^\dagger a_{\boldsymbol{k}} + \frac{1}{2}) - \frac{N_T \lambda}{2} + E_0, \qquad (30)$$

we obtain the ground state energy $E$ as

$$E = \frac{1}{2} \sum_{\boldsymbol{k} \in \mathrm{B.Z.}} \omega_{\boldsymbol{k}} - \frac{N_T \lambda}{2} + E_0, \qquad (31)$$

where $a_{\boldsymbol{k}}$ is a linear combination of $b_{\boldsymbol{k}}$ and $b_{\boldsymbol{k}}^\dagger$ via the Bogoliubov transformation, and the spinon dispersion is given as

$$\omega_{\boldsymbol{k}} = (\lambda^2 - 2h_x \mathcal{B} \lambda \gamma_{\boldsymbol{k}})^{1/2}. \qquad (32)$$

Here $\gamma_{\boldsymbol{k}} = 2[\cos(\sqrt{3}k_x)\cos(k_y) + \cos(2k_y)]$. We have set the lattice constant of the kagome lattice to 1.

The mean-field parameters $\lambda$ and $\mathcal{B}$ are determined by solving the self-consistent equations [23],

$$\partial E/\partial \lambda = 0, \quad \partial E/\partial \mathcal{B} = 0. \qquad (33)$$

Since $\mathcal{B}$ minimizes the ground state energy $E$, we must also ensure that $\partial^2 E/\partial \mathcal{B}^2 > 0$. Furthermore, as $b_{\boldsymbol{r}}^\dagger$ and $B_{\boldsymbol{rr}'}^\dagger$ represent hardcore bosons, the mean-field value $\mathcal{B}$ must satisfy the conditions $0 \leq \mathcal{B}^2 \leq 1$ and $0 \leq Q - 6\mathcal{B}^2 \leq 1$.

To prevent the spinon condensation, the spinon dispersion $\omega_{\boldsymbol{k}}$ should have a finite energy gap. By solving the self-consistent equations, we identify several regions in the parameter space $(\mathcal{V}, h_z, h_x)$ where the spinons remain stable. Each region corresponds to a distinct type of $\mathbb{Z}_2$ spin liquid. Specifically, we find four types of $\mathbb{Z}_2$ spin liquid [23], corresponding to $Q = 1, 2$ and $\mathcal{B} < 0, \mathcal{B} > 0$ (See Fig. 3 for further details). At the phase boundary of $\mathbb{Z}_2$ spin liquids, the minimum of the spinon energy band touches zero, the system is then condensed to ordered phases. The mean-field calculation here is mainly focused on the possible spin liquid phases, in which gauge fields are condensed. In reality, there might be other competing ordered phases such as the paramagnetic phase, staggered phase, nematic phase, stripe phase, and string phase [24].

## VI. EXCITATIONS AND SPECTRUM

In this section, we focus on the $\mathbb{Z}_2$ topological ordered phases of our model and explore the spectroscopic properties of the fractionalized excitations. For the $\mathbb{Z}_2$ topological order, the fractional excitations are spinons and visons [23, 39, 43]. They are sometimes referred to as $e$ particles and $m$ particles (see Fig. 2). Here we do not talk about the composite of $e$ and $m$ as it is not directly manifested in the pair-wise spin correlators.

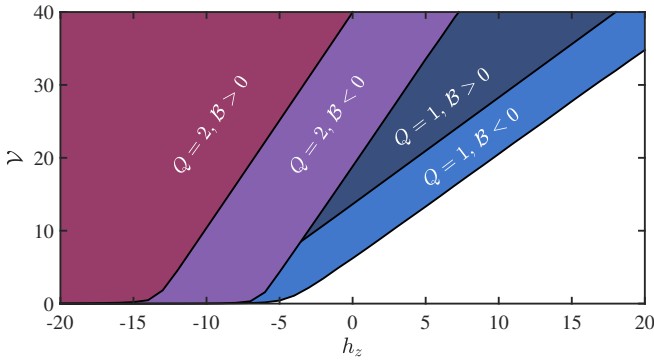

FIG. 3. The phase boundaries of possible spin liquid solutions are mapped in the parameter space $(h_z, \mathcal{V})$, with the transverse field $h_x = 1$ taken as the energy unit. Each colored region represents a distinct spin liquid phase, while the black curves indicate the phase boundaries between these spin liquid phases and non-liquid phases. Some regions exhibit overlap between different spin liquid phases. The phases with $Q = 1$ ($Q = 2$) correspond to odd (even) $\mathbb{Z}_2$ spin liquids, and the sign of $\mathcal{B}$ determines the minimum of the spinon dispersion.

In our model, spinons are created by flipping the $S^z$ spin. Thus, the spinons are excitations by breaking the dimers.

In contrast, visons are created by modulating the superposition of the dimer coverings without breaking the dimers. The energy scale of visons is much smaller compared to the one of spinons. The spectra of spinons and visons are well separated, and we will compute them explicitly below.

### A. Spinon continuum

The spinon operators $\tau_{\boldsymbol{r}}^z$ are related to the physical spin operator $S_i^x$ via $S_i^x = \frac{1}{2}\sigma_{\boldsymbol{rr}'}^z \tau_{\boldsymbol{r}}^z \tau_{\boldsymbol{r}'}^z$, and incorporates the spinon creation and annihilation operators $a_{\boldsymbol{k}}$ and $a_{\boldsymbol{k}}^\dagger$ under the hardcore boson representation. Thus, the spin correlator $\langle S_i^x(t)S_i^x(0)\rangle$ that is measured by the neutron scattering experiments, includes the contribution from the spinon continuum. Assuming a uniform gauge field condensation, $\langle S_i^x(t)S_i^x(0)\rangle$ is proportional to the density of states of spinon continuum $\rho_{\mathrm{spinon}}(\boldsymbol{k}, E)$ up to a nonuniversal form factor [69]. Moreover, according to the energy-momentum conservation, the two spinons share the energy and momentum loss $(E, \boldsymbol{k})$ of the neutron as $E = \omega_{\boldsymbol{q}_2} + \omega_{\boldsymbol{q}_2}$ and $\boldsymbol{k} = \boldsymbol{q}_1 + \boldsymbol{q}_2$, where $\omega_{\boldsymbol{q}}$ is the spinon dispersion. The structure of the spinon continuum in the energy and the momentum domains are constrained by the spinon dispersion.

The odd and even $\mathbb{Z}_2$ gauge theory description for the $\mathbb{Z}_2$ topological order differs in the flux that is experienced by the vison, and does not occur in the spinon sector.

Thus, the spinon continuum for the odd and even $\mathbb{Z}_2$ topological orders has no qualitative difference. In Fig. 4, we depict the spinon continuum of two examples of the

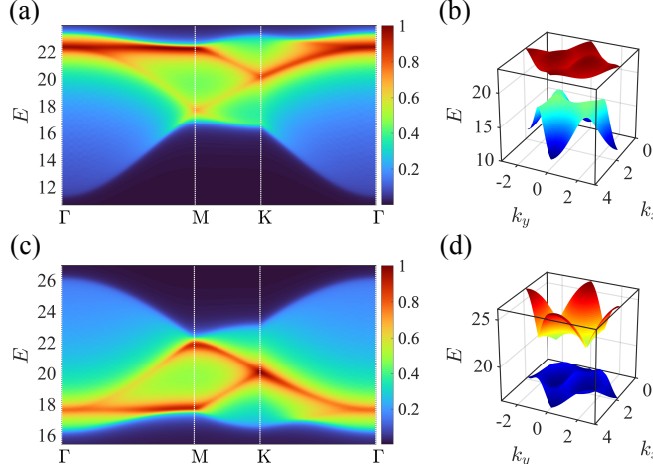

FIG. 4. The spinon continuum of the even $\mathbb{Z}_2$ spin liquid is shown along the high-symmetry lines in (a) for positive $\mathcal{B}$; in (c) for negative $\mathcal{B}$. The lower and upper limits of the continuum in the first Brillouin zone for both cases are plotted in (b) and (d), respectively. The model parameters for this figure are $\mathcal{V} = 30$ and $h_z = -15$, which correspond to $\lambda = 10.07$ and $\mathcal{B} = 0.5762$ for $\mathcal{B} > 0$ case; $\lambda = 10.05$ and $\mathcal{B} = -0.5767$ for $\mathcal{B} < 0$ case. For the odd $\mathbb{Z}_2$ quantum spin liquid, the spinon continuum has no qualitative difference then the even one, hence is not shown in the figure.

$\mathbb{Z}_2$ topological orders, $\mathcal{B} > 0$ and $\mathcal{B} < 0$. It most obvious features of the continuum are the reverse of low-energy and high-energy parts. For $\mathcal{B} > 0$, the minima of the spinon energy band are located at two inequivalent K points. The condensation of two spinons at K points leads to either an ordered phase with a wave vector K or a uniform phase with wave vector Γ. For $\mathcal{B} < 0$, spinons feel a $\pi$ flux around the triangular plaquettes of the triangular lattice. Such a flux pattern in the triangular lattice doesn't cause symmetry fractionalization but reverses the energy band. Now, the minimum of the spinon energy band occurs at the Γ point. Thus, the minimum of the spinon continuum is also located at the Γ point, leading to a uniform phase when condensing. The spinon continuum minima can be used to differentiate two kinds of $\mathbb{Z}_2$ spin liquids in experiments.

In the analysis of spinon mean-field theory, we have ignored the quantum dynamics generated from the perpendicular spin-flipping $J_{\pm}$-term. Actually, the $J_{\pm}$-term leads to the second-nearest hopping of spinons, which changes the shape of the spinon spectrum. In the presence of a nonzero $J_{\pm}$, the spinon continuums for $B > 0$ and $B < 0$ show more different structures, depending on the relative strength between $J_{\pm}$ and $h_x$ (see the Appendix).

### B. Vison continuum

#### 1. Duality transformation

To reveal the vison excitations, the $\mathbb{Z}_2$ lattice gauge theory in Eq. (19) on the triangular lattice is further transformed to the dual $\mathbb{Z}_2$ gauge theory on the dual honeycomb lattice (see Fig. 2) [23, 42, 50, 70]. We denote the honeycomb lattice sites as $\boldsymbol{R}$. At the sites of the honeycomb lattice, we define a dual $\mu$ matter field; at the links, we introduce a dual $\eta$ gauge field. Due to the one-to-one mapping between (i) the links of the triangular lattice and the links of the honeycomb lattice, (ii) the sites of the triangular lattice and the plaquettes of the honeycomb lattice, and (iii) the plaquettes of the triangular lattice and the sites of the honeycomb lattice, we can associate $\sigma$ field and $\tau$ field with $\eta$ field and $\mu$ field,

$$
\tau_{\boldsymbol{r}}^x = \prod_{\boldsymbol{R}\boldsymbol{R}' \in \bigcirc_{\boldsymbol{r}}} \eta_{\boldsymbol{R}\boldsymbol{R}'}^z, \quad \mu_{\boldsymbol{R}}^x = \prod_{\boldsymbol{R}\boldsymbol{R}' \in \triangle_{\boldsymbol{R}}} \sigma_{\boldsymbol{R}\boldsymbol{R}'}^z,
$$
$$
\sigma_{\boldsymbol{R}\boldsymbol{R}'}^x = \eta_{\boldsymbol{R}\boldsymbol{R}'}^z \mu_{\boldsymbol{R}}^z \mu_{\boldsymbol{R}'}^z, \quad \eta_{\boldsymbol{R}\boldsymbol{R}'}^x = \sigma_{\boldsymbol{R}\boldsymbol{R}'}^z \tau_{\boldsymbol{r}}^z \tau_{\boldsymbol{r}'}^z. \tag{34}
$$

In the second row, the triangular lattice link $\boldsymbol{R}\boldsymbol{R}'$ has the same center as the honeycomb lattice link $\boldsymbol{R}\boldsymbol{R}'$. Substituting these relations into the $\mathbb{Z}_2$ gauge theory Eq. (19), we obtain the dual gauge theory

$$
H = \frac{1}{2} \sum_{\boldsymbol{R}_1\boldsymbol{R}_2, \boldsymbol{R}_3\boldsymbol{R}_4}^{\boldsymbol{R}_1\boldsymbol{R}_2 \neq \boldsymbol{R}_3\boldsymbol{R}_4} \frac{V_{\boldsymbol{R}_1\boldsymbol{R}_2, \boldsymbol{R}_3\boldsymbol{R}_4}}{4} \eta_{\boldsymbol{R}_1\boldsymbol{R}_2}^z \eta_{\boldsymbol{R}_3\boldsymbol{R}_4}^z
$$
$$
\times \mu_{\boldsymbol{R}_1}^z \mu_{\boldsymbol{R}_2}^z \mu_{\boldsymbol{R}_3}^z \mu_{\boldsymbol{R}_4}^z - \frac{h_z}{2} \sum_{\boldsymbol{R}\boldsymbol{R}'} \eta_{\boldsymbol{R}\boldsymbol{R}'}^z \mu_{\boldsymbol{R}}^z \mu_{\boldsymbol{R}'}^z \tag{35}
$$
$$
- \frac{h_x}{2} \sum_{\boldsymbol{R}\boldsymbol{R}'} \eta_{\boldsymbol{R}\boldsymbol{R}'}^x,
$$

where $V_{\boldsymbol{R}_1\boldsymbol{R}_2, \boldsymbol{R}_3\boldsymbol{R}_4}$ is also a renaming of the dipolar interaction $V_{ij}$. The gauge constraint Eq. (20) is automatically satisfied by the duality transformation.

Tracking the duality transformation, one can find $S_i^z = \eta_{\boldsymbol{R}\boldsymbol{R}'}^z \mu_{\boldsymbol{R}}^z \mu_{\boldsymbol{R}'}^z$. The spin liquid ground state is a superposition of many states with different $S^z$ textures. The action of $S_i^z$ on the ground states changes the superposition coefficients, creating two visons at the honeycomb lattice sites $\boldsymbol{R}$ and $\boldsymbol{R}'$. Thus, $\mu_{\boldsymbol{R}}^z$ is interpreted as the vison creation operator, while $\mu_{\boldsymbol{R}}^x$ records the vison number.

#### 2. Vison mean-field Hamiltonian

When a spin liquid is stabilized, it is also possible to discuss the dynamics of visons in the background of spinon matter. Visons and Spinons obey mutual semion statistics. Such a property can be quantitatively described by a Wilson loop operator. In $\mathbb{Z}_2$ spin liquid ground state $|\Psi\rangle$, the Wilson loop of $\sigma$ gauge field has

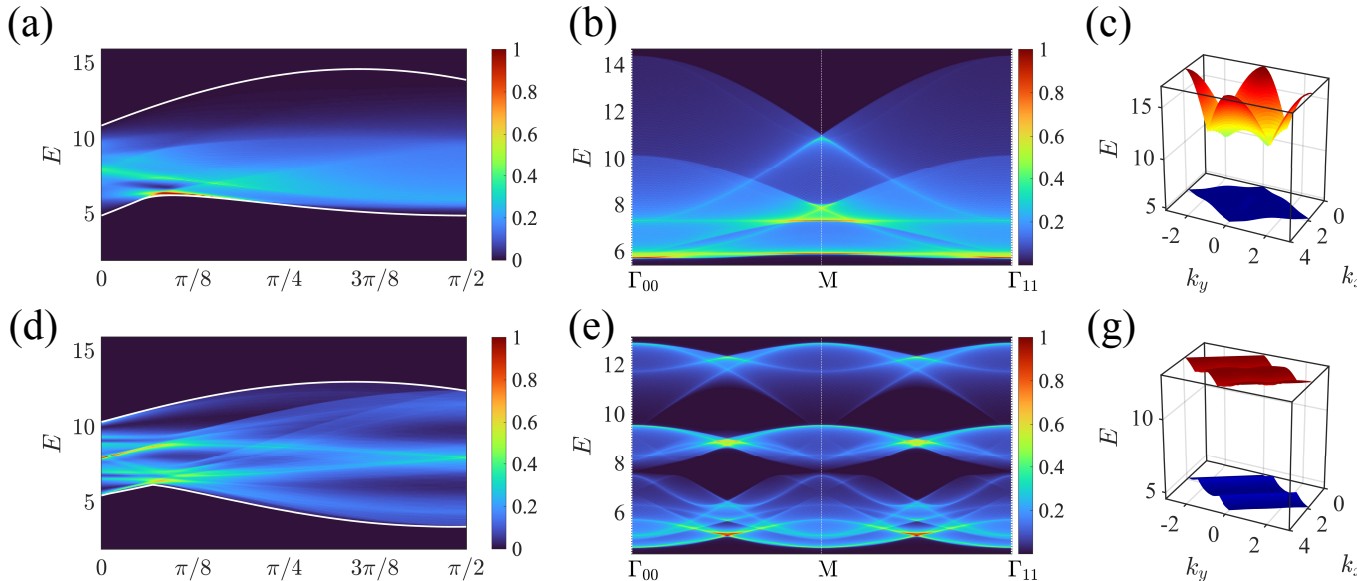

FIG. 5. The density of states $\rho_{\text{vison}}^{\text{even/odd}}(\boldsymbol{k}, E)$ (normalized by its maximum) of vison continuum for even and odd $\mathbb{Z}_2$ quantum spin liquids. Model parameters are $\mu = 4$, $J_1 = \cos\phi$ and $J_2 = \sin\phi$. The three figures in the upper panel are for even $\mathbb{Z}_2$, while these in the lower panel are for odd $\mathbb{Z}_2$. (a) and (d) show the integrated density of states $\rho_{\text{vison}}^{\text{even/odd}}(E) = \sum_{\boldsymbol{k}} \rho_{\text{vison}}^{\text{even/odd}}(\boldsymbol{k}, E)$ as a function of $\phi$. The boundaries of the continuum are indicated by white solid lines. (b) and (e) are the plot of $\rho_{\text{vison}}^{\text{even/odd}}(\boldsymbol{k}, E)$ along the high-symmetry lines by fixing $\phi = \pi/4$. The enhanced periodicity of odd $\mathbb{Z}_2$ is clear in (e). (c) and (g) plot the upper limit and lower limit of the vison continuum for $\phi = \pi/4$.

determined value

$$\prod_{\boldsymbol{R}\boldsymbol{R}' \in \curlywedge_r} \sigma_{\boldsymbol{R}\boldsymbol{R}'}^x |\Psi\rangle = W |\Psi\rangle, \tag{36}$$

where $W = \pm 1$. The value $W$ should be uniform for all the hexagons in the kagome lattice because $\mathbb{Z}_2$ spin liquid does not break any translational symmetry. We call $|\Psi\rangle$ an even (odd) $\mathbb{Z}_2$ spin liquid state if $W = 1$ ($W = -1$). Gauge constraint relates the spinon number operator $\tau_i^x$ with the gauge charge. Therefore, in the even $\mathbb{Z}_2$ spin liquid, there is no spinon distributed in the triangular lattice, while in the odd $\mathbb{Z}_2$ spin liquid, there is one spinon per site of the triangular lattice. When a vison goes around a spinon, it feels a $\pi$ flux. Such an effect manifests itself in the gauge field $\eta$. One can derive from the duality relations Eq. (34) that the $\eta$ gauge field satisfies $\prod_{\boldsymbol{R}\boldsymbol{R}' \in \bigcirc_r} \eta_{\boldsymbol{R}\boldsymbol{R}'}^z = W$ for two kinds of $\mathbb{Z}_2$ spin liquid states.

To explore the dynamics of visons, we treat the spin liquid ground state as a mean field, in which the $\eta^z$ gauge field is locked to a mean-field value $\eta_{\boldsymbol{R}\boldsymbol{R}'}^z = \bar{\eta}_{\boldsymbol{R}\boldsymbol{R}'}^z$ that generates the 0 or $\pi$ flux for the even or odd spin liquids. Then the low-energy physics of Eq. (35) is described by a mean-field Hamiltonian

$$\mathcal{H}_{\text{vison}} = \frac{1}{2} \sum_{\boldsymbol{R}_1\boldsymbol{R}_2, \boldsymbol{R}_3\boldsymbol{R}_4}^{\boldsymbol{R}_1\boldsymbol{R}_2 \neq \boldsymbol{R}_3\boldsymbol{R}_4} \frac{V_{\boldsymbol{R}_1\boldsymbol{R}_2, \boldsymbol{R}_3\boldsymbol{R}_4}}{4} \bar{\eta}_{\boldsymbol{R}_1\boldsymbol{R}_2}^z \bar{\eta}_{\boldsymbol{R}_3\boldsymbol{R}_4}^z$$
$$\times \mu_{\boldsymbol{R}_1}^z \mu_{\boldsymbol{R}_2}^z \mu_{\boldsymbol{R}_3}^z \mu_{\boldsymbol{R}_4}^z - \frac{h_z}{2} \sum_{\boldsymbol{R}\boldsymbol{R}'} \bar{\eta}_{\boldsymbol{R}\boldsymbol{R}'}^z \mu_{\boldsymbol{R}}^z \mu_{\boldsymbol{R}'}^z. \tag{37}$$

The out-of-plane Zeeman field $h_z$ gives rise to nearest-neighbor vison interaction, while the dipolar interaction $V_{\boldsymbol{R}_1\boldsymbol{R}_2, \boldsymbol{R}_3\boldsymbol{R}_4}$ is associated with four vison operators and leads to long-range interaction. If we truncate $V_{\boldsymbol{R}_1\boldsymbol{R}_2, \boldsymbol{R}_3\boldsymbol{R}_4}$ at the nearest neighbor, the links $\boldsymbol{R}_1\boldsymbol{R}_2$ and $\boldsymbol{R}_3\boldsymbol{R}_4$ share a common site. The four vison operators are then reduced to two vison operators. The resulting Hamiltonian becomes an Ising model with nearest-neighbor interaction $J_1 = -h_z/2$ and next-nearest-neighbor interactions $J_2 = V_1/4$,

$$\mathcal{H}_{\text{vison}} = J_1 \sum_{\boldsymbol{R}\boldsymbol{R}'} \bar{\eta}_{\boldsymbol{R}\boldsymbol{R}'}^z \mu_{\boldsymbol{R}}^z \mu_{\boldsymbol{R}'}^z$$
$$+ J_2 \sum_{\boldsymbol{R}\boldsymbol{R}'} \bar{\eta}_{\boldsymbol{R}\boldsymbol{R}''}^z \bar{\eta}_{\boldsymbol{R}''\boldsymbol{R}'}^z \mu_{\boldsymbol{R}}^z \mu_{\boldsymbol{R}'}^z, \tag{38}$$

where $\boldsymbol{R}''$ is the honeycomb lattice site that makes $\boldsymbol{R}\boldsymbol{R}''$ and $\boldsymbol{R}''\boldsymbol{R}'$ connect. The truncation of dipolar interaction at $V_1$ is justified for the restricted BFG model where $V_1 \gg V_{n\geq 2}$. For the conventional BFG model where $V_1$, $V_2$, and $V_3$ have close strength, one can treat the high-order dipolar interactions as a mean field and give rise to the renormalization of $J_1$ and $J_2$. Under this approximation, it is possible to work out the vison dispersion, which is nothing but the soft modes of the Ising Hamiltonian Eq. (38).

### 3. Even $\mathbb{Z}_2$ spin liquids

For the even $\mathbb{Z}_2$ spin liquid, each hexagon of the honeycomb lattice contains no gauge flux, i.e., $\prod_{\boldsymbol{RR'}\in\bigcirc_r}\bar{\eta}^z_{\boldsymbol{RR'}}=1$. In terms of X-G Wen's scheme of classification, this belongs to a Z2A spin liquid for the vison sector [46]. We can trivialize the mean gauge field as $\bar{\eta}^z_{\boldsymbol{RR'}}=1$. Diagonalizing Eq. (38), we obtain the vison dispersion

$$\epsilon^{\text{even}}_{\pm,\boldsymbol{k}}=\mu+\frac{J_2}{2}\gamma_{\boldsymbol{k}}\pm\frac{|J_1|}{2}|\zeta_{\boldsymbol{k}}|, \qquad (39)$$

where $\gamma_{\boldsymbol{k}}=2[\cos(\sqrt{3}k_x)\cos(k_y)+\cos(2k_y)]$ and $\zeta_{\boldsymbol{k}}=1+2e^{i\sqrt{3}k_x}\cos(k_y)$. Notably, the $J_2$ interaction of the honeycomb lattice forms two decoupled triangular lattices, and $\zeta_{\boldsymbol{k}}$ and $\gamma_{\boldsymbol{k}}$ are the structure factors of the honeycomb lattice and the triangular lattice, respectively. We have added a chemical potential $\mu$ to avoid the vison condensation.

The vison dispersion shows a competition between the triangular lattice part and the honeycomb lattice part. When $J_2\gg|J_1|$, $\gamma_{\boldsymbol{k}}$ dominates, the dispersion looks like the one of a triangular lattice. In this case, the energy minimum is at the K points of the Brillouin zone. When $|J_1|\gg J_2$, $|\zeta_{\boldsymbol{k}}|$ dominates and the dispersion now looks like the one of a honeycomb lattice, which has two inequivalent Dirac cones at two K points. The energy minimum moves to the $\Gamma$ point. Thus, the vison condensation of these two cases may cause the Ising ordered phases with different magnetic wave vectors. In the intermediate regime, the vison minima actually develop the contour degeneracies in the reciprocal space, which is quite similar to the spin correlation for the honeycomb lattice classical spiral spin liquid [71–73]. The Ising nature of the vison field may bring extra complication to these contour degeneracy [72]. This contour degeneracy is not protected by symmetry, and the vison interaction and further neighbor vison hoppings should be able to lift the degeneracy and lead to other competing Ising orders once the vison is condensed [23, 43].

When the visons are deconfined in the spin liquid phase, the spin correlation function $S^{zz}_{ij}(t)=\langle S^z_i(t)S^z_j(0)\rangle$ contains the signal of the vison continuum, because $2S^z_i=\sigma^x_{\boldsymbol{RR'}}=\eta^z_{\boldsymbol{RR'}}\mu^z_{\boldsymbol{R}}\mu^z_{\boldsymbol{R'}}$ is related to two vison fields. According to the energy-momentum conservation, the DoS of the vison continuum $\rho^{\text{even}}_{\text{vison}}(\boldsymbol{k},E)$ is given by $E=\epsilon^{\text{even}}_{\alpha,\boldsymbol{q}_1}+\epsilon^{\text{even}}_{\beta,\boldsymbol{q}_2}$ and $\boldsymbol{k}=\boldsymbol{q}_1+\boldsymbol{q}_2$, where $\alpha$ and $\beta$ run in two vison bands.

### 4. Odd $\mathbb{Z}_2$ spin liquids

For the odd $\mathbb{Z}_2$ spin liquid, each hexagon contains a $\pi$ gauge flux, $\prod_{\boldsymbol{RR'}\in\bigcirc_r}\bar{\eta}^z_{\boldsymbol{RR'}}=-1$. In terms of X-G Wen's scheme of classification, this belongs to a Z2B spin liquid for the vison sector. We fix the gauge by $\bar{\eta}^z_{\boldsymbol{RR'}}=-\exp(i\xi_{\boldsymbol{RR'}}\boldsymbol{Q}\cdot\boldsymbol{R})$, where $\xi_{\boldsymbol{RR'}}=1$ for those

links that are parallel to $x$-direction, and 0 for the others. The wave vector $\boldsymbol{Q}=\pi(-\frac{1}{2\sqrt{3}},\frac{1}{2})$ accounts for the periodicity of the gauge field. Due to the $\pi$ flux, the unit cell of the honeycomb lattice is enlarged to the magnetic unit cell, and the translational symmetry is fractionalized. Such an effect leads to the enhanced periodicity of the vison spectrum. What is interesting here is that the translational symmetry of the spin liquid ground state is not broken, but the periodicity of vison spectrum is enhanced.

The DoS $\rho^{\text{odd}}_{\text{vison}}(\boldsymbol{k},E)$ of the vison continuum for the odd $\mathbb{Z}_2$ spin liquids is given by $E=\epsilon_{\alpha,\boldsymbol{q}_1}+\epsilon_{\beta,\boldsymbol{q}_2}$ and $\boldsymbol{k}=\boldsymbol{q}_1+\boldsymbol{q}_2+\boldsymbol{Q}$, in which the wave vector of the nonuniform gauge field $\bar{\eta}^z$ plays the role of a momentum offset. In Fig. 5, we compare the vison continuum of even and odd $\mathbb{Z}_2$ spin liquids. The enhanced periodicity of the odd $\mathbb{Z}_2$ spin liquid is depicted in Fig. 5(e).

## VII. DISCUSSION

We have systematically studied the $\mathbb{Z}_2$ topological orders for the kagome dipolar systems, from the microscopic degrees of freedom and the model construction to the perturbative analysis and the non-perturbative mean-field theory. The perturbative treatment and non-perturbative treatment are complementary to each other and provide useful insights to understand the emergent $\mathbb{Z}_2$ topological orders in the system. The dynamic properties of the $\mathbb{Z}_2$ topological orders for the spinon and vison sectors are then studied in different regimes. In the following, we explain the experimental consequences and detection of the $\mathbb{Z}_2$ topological orders for the dipolar kagome systems and then discuss the material relevance and polar molecules.

### A. Experimental consequences and detection of $\mathbb{Z}_2$ topological order

Fundamentally, the $\mathbb{Z}_2$ topological order belongs to the family of string-net condensed states [63, 64, 74]. In the string-net description, the topological order corresponds to the condensation of the closed strings, while the ends of open strings correspond to the emergent fractionalization quasiparticles/anyons. In the $\mathbb{Z}_2$ topological ordered state, these strings have no tension, and thus the end particles are deconfined. For our specific case, the $S^x_i$ breaks the dimer and thus creates the shortest open strings whose ends are two spinons. On the opposite, when the strings have a finite tension, these particles are confined. The recent technology of the Rydberg atom array enables the direct exploration of confinement and string breaking dynamics with a high spatiotemporal resolution by quenching certain local controllable parameters in the quantum simulation [75], in which rather than a $\mathbb{Z}_2$ lattice gauge theory a U(1) one is constructed when truncating at the strongest $V_1$ interaction [28, 75]. Here

we focus on the properties of $\mathbb{Z}_2$ spin liquids. The solid-state-based experiments have not yet been able to access such spatio-temporal dynamics of the strings. Instead, we discuss the more traditional thermodynamic and spectroscopic measurements for the kagome dipolar magnets [76].

### 1. non-Kramers doublets

We start with the thermodynamics. The simple entropy measurement, that is obtained from the specific heat, could reflect the interaction and correlation of the system [53, 76]. Let us consider the restricted BFG case. If one cools the system from the high-temperature paramagnetic phase, the entropy drops from $S = R\ln 2$ per spin to an entropy plateau at a temperature scale $T \sim \mathcal{O}(V_1)$. This entropy plateau is the kagome spin ice type of entropy that counts the degeneracy of the Ising spin configuration on the triangular plaquette. Using Pauling's argument, this entropy is $S \sim (1/3)R\ln(9/2) \approx 0.50R$ per spin [77, 78]. If one further cools the system down to the temperature of the scale of $T \sim \mathcal{O}(V_2, V_3)$, this reduces the degeneracy and the entropy plateau occurs at $S_{\mathrm{rBFG}}$ that is $\approx 0.28R$ per spin (estimated for a kagome lattice with $2 \times 3$ unit cells and periodic boundary conditions). As the temperature goes further down, the (quantum mechanical) "dimer resonating" process starts to play a role, and the system is moving towards the actual ground state. For the conventional BFG case, there is no such two-entropy-plateau process. Since $V_1, V_2$ and $V_3$ are close in energy, the system reaches the entropy plateau with $S \sim (1/3)R\ln(5/2) \approx 0.31R$ per spin at the temperature scale of $\mathcal{O}(V_1, V_2, V_3)$. As the "dimer resonating" process starts to play a role at even lower temperatures, the system moves toward the ground state. The multi-step entropy behaviors are shown in Fig. 6.

For the $\mathbb{Z}_2$ topological order, all the excitations are gapped [46]. At low temperatures, the specific heat of the system should have a gapped behavior [79, 80]. With an independent vison gas and spinon gas assumption, the specific heat is expected to have a dependence on the temperature like

$$C_v(T) \sim c_1 e^{-\Delta_{\mathrm{m}}/T} + c_2 e^{-\Delta_{\mathrm{s}}/T}, \qquad (40)$$

where $c_1, c_2$ are prefactors, and $\Delta_{\mathrm{m}}$ and $\Delta_{\mathrm{s}}$ are the vison gap and the spinon gap, respectively. Due to the separation of the energy scales between the spinon and the vison, the very low-temperature specific heat should be mostly contributed by the activated visons. Owing to the vison and spin gas in the thermodynamic measurement, the gaps extracted from the thermodynamics are the single vison gap and the single spinon gap. Likewise, the low-temperature spin susceptibility also reveals such gaps. For the non-Kramers doublet, since only the $S^z$ is coupled to the magnetic field and $S^z$ creates the vison pairs, thus only the vison gap is revealed in the spin

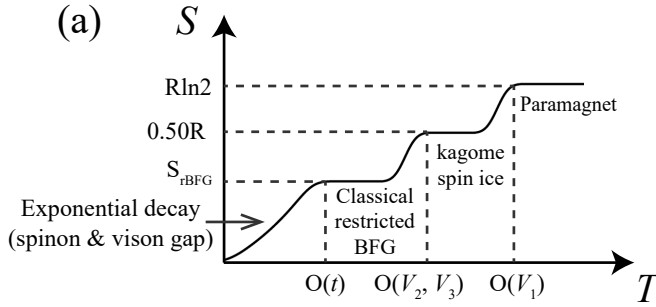

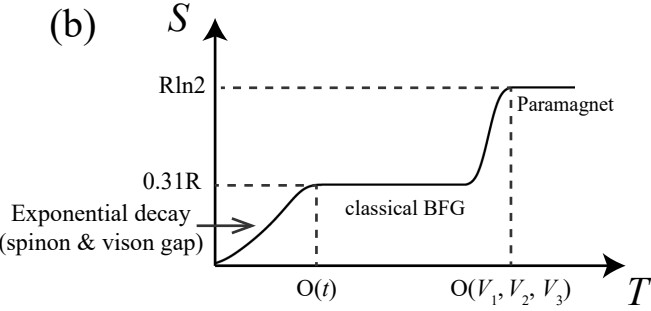

FIG. 6. Entropy $S$ of kagome dipolar magnets shows several plateaux as the temperature $T$ decreases. (a) is for the restricted BFG case ($V_1 \gg V_2 \sim V_3$). At high-temperature, the system has a paramagnetic entropy $R\ln 2$. When $T$ is near the energy scale $O(V_1)$, the kagome spin ice entropy $\sim 0.50R$ starts to appear. Further decreasing $T$ to the energy scale $O(V_2, V_3)$, the entropy goes to the classical entropy of the restricted BFG model, which is estimated as $S_{\mathrm{rBFG}} \approx 0.28R$ from a kagome lattice with $2 \times 3$ unit cells. When the temperature is lower than the energy scale of the "dimer resonating" process $O(t)$, the effects of spin liquids appear, which make $S$ decay exponential with spin gap $\Delta_{\mathrm{s}}$ and vison gap $\Delta_{\mathrm{m}}$. (b) is for the normal BFG case ($V_1 \sim V_2 \sim V_3$). The decreasing of $T$ from the paramagnetic scale to $O(V_1, V_2, V_3)$ makes $S$ drop to the plateau of the classical BFG model. Further decreasing $T$ also leads to exponential decay due to the spin liquids.

susceptibility as

$$\chi(T) \sim \chi_0 + c_3 e^{-\Delta_{\mathrm{m}}/T}, \qquad (41)$$

where $c_3$ is a prefactor, and $\chi_0$ is the zero-temperature susceptibility. For the Kramers doublets, the spinon and vison contributions can be separately identified by considering the *probing* fields along in-plane and out-of-plane directions for our model, respectively [50]. The spin susceptibility is better measured by the Knight shift in an NMR or $\mu$SR experiment [81, 82].

In contrast to the thermodynamic results, the spectroscopic measurements provide more useful information about the emergent quasiparticles in the system [54, 76, 83, 84]. This can be achieved with the inelastic neutron scattering measurement and the $1/T_1$ spin-lattice relaxation time measurement of NMR [85]. More crucially, there exists an interesting and important selective measurement for the visons and the spinons. This selective measurement of emergent quasiparticles was originally proposed for the non-Kramers doublets and the

dipole-octupole doublets in the context of pyrochlore quantum spin ice [50, 86] and is found to be quite useful in the current context. We start with the non-Kramers doublet. For the non-Kramers doublet, as only $S^z$ is time-reversal odd, the $S^z$-$S^z$ correlation is automatically selected in the neutron scattering or $1/T_1$ NMR measurements. Thus, the vison continuum is naturally selected in these measurements. In particular, the spectral gap, which is recorded in these spectroscopic measurements, is two vison gaps, and is *twice* of the vison gap in the thermodynamic measurement. The inelastic neutron scattering provides more information about the momentum distribution. If the vison experiences a background flux and thus the crystal symmetry fractionalization, the vison continuum, which is detected by the neutron, could exhibit the enhanced spectral periodicity in the reciprocal space (see Fig. 5). The background flux also creates multiple vison bands, and each vison band has a relatively narrower bandwidth than the zero-flux case. The vison continuum further exhibits more peak structures in the energy domain (see Fig. 5). Besides the measurement of a continuum in INS experiments, it is also possible to detect the mutual statistics between spinons and visons by threshold threshold spectroscopy experiments [87–89], which reveal dynamical signals of fractional excitations. These are useful signatures to be experimentally examined.

### 2. Kramers doublets

For the Kramers doublet, the spin correlations of all spin components are included in the spectroscopic measurements. One could use the polarized neutrons to separate the $z$-component and the in-plane component correlations. The former corresponds to the vison continuum, and the latter corresponds to the spinon continuum. Nevertheless, deeply inside the $\mathbb{Z}_2$ topological ordered phase rather than the phase boundary, the spinon has a much higher energy scale than the vison, and in principle, one can visualize the spinon continuum and the vison continuum even in the energy domain. Since the structures of the vison continuum have been explained, we turn to the structures of the spinon continuum. Again, the gap of the spinon continuum is twice the spinon gap that is extracted from the thermodynamics.

The spinon also experiences a background 0 flux ($\mathcal{B} > 0$) or a $\pi$ flux ($\mathcal{B} < 0$) generated by the visons. However, $\pi$ flux does not cause symmetry fractionalization in the triangular lattice [90]. The periodicity of the spinon continuum is not enhanced (see Fig. 4). In fact, the background flux reverses the minima and maxima of the continuum. For the 0 flux case, the minima occur at $\Gamma$ point and K points; while for the $\pi$ flux case, the minimum is at the $\Gamma$ point. Moreover, the high-density regime of the continuum is crowded near the lower or upper limits for the 0 flux or $\pi$ flux case, respectively. These features of the spinon continuum can be used to differentiate the type of spin liquids in the measurements.

### 3. Cluster Mott insulators

We turn to the cluster Mott insulator with the electron degrees of freedom and describe the difference from the spin degrees of freedom. We only describe the 1/4-filling case, and other fillings can be understood in the similar manner. It is convenient to convey the discussion with an extended Hubbard model on the kagome lattice with

$$H = \sum_{ij} \left( -t_{ij} d_{i\alpha}^\dagger d_{j\alpha} + V_{ij} n_i n_j \right) + U \sum_i n_{i\uparrow} n_{i\downarrow}, \quad (42)$$

where $d_{i\alpha}^\dagger$ ($d_{i\alpha}$) creates (annihilates) the electron with spin $\alpha$ at the site $i$, $U$ is the usual onsite interaction, and $V_{ij}$ is the neighboring repulsive interaction with $V_1, V_2, V_3$ the first, second, and third neighbor interaction on the hexagon plaquette, respectively. Here $V_{ij}$'s arise from the Coulomb interaction and depend on the relevant orbital content. The Hubbard-$U$ alone cannot localize the electron and create a Mott state. Being a large energy scale, the role of $U$ is simply to suppress the double occupation. It is the $V_{ij}$'s that localize the electrons and create the cluster Mott insulating states.

For the restricted BFG regime of Fig. 1, the electron is first localized in *each* triangular plaquette at the energy/temperature scale of $V_1$, and the electron number of each triangular plaquette is restricted to 1 or 2. The electron correlation on the hexagons enters at the energy scale of $V_2$ and $V_3$, such that the total number of electrons on the hexagon is then restricted to 3. For the BFG regime of Fig. 1, $V_1$ and $V_2, V_3$ are close in energy, and the electron is localized on the hexagon plaquette. These clusterly localized electrons further develop a correlated motion on the hexagon plaquette, which is equivalent to the dimer resonating process of Sec. IV. As a result, the charge sector could develop a $\mathbb{Z}_2$ topological order.

One immediate consequence of the charge-sector $\mathbb{Z}_2$ topological order is the charge fractionalization [91–93]. The fractionalized chargeon carries $1/2$ of the electron charge. The charge fractionalization can be detected via the shot-noise measurement [94, 95]. The chargeon continuum can be recorded in the electron spectral function. The complication here is the fate of the spin of the electron which may also be exotic on its own and form a quantum spin liquid. On the other hand, the electron density-density correlation, which can be measured by X-ray scattering [96–98], encodes the vison continuum of the $\mathbb{Z}_2$ topological order.

### 4. Polar molecules

For polar molecules, the dipole moments come from the rotational degrees of freedom of the two-atom molecules. One important difference between polar molecules and

non-Kramers or Kramers doublets is that the dipole moments of polar molecules are electric rather than magnetic. Namely, instead of being coupled with the magnetic field, polar molecules are coupled with the electric field. These differences lead to different experimental techniques of measurement. Nevertheless, the thermodynamics results are still valid, though the thermodynamic measurement in the ultracold atom systems can be quite challenging. Instead, one often chooses to perform certain correlation measurements to extract the spectroscopic properties. To access the spectroscopic properties of the $\mathbb{Z}_2$ liquid phase in the polar molecule systems, one can use the two-photon Raman spectroscopy [99] to obtain equivalent outcomes as the inelastic neutron scatterings for the spins. The polarization degrees of freedom of photons can be coupled with the electric dipole moment, hence the two-photon Raman spectroscopy measures the correlation function of the electric dipole moment, which contains the spinon and vison continuum in the liquid phase. Moreover, it is also possible to use the similar technology of the Rydberg atom array with a high spatiotemporal resolution to explore the string dynamics in polar molecule systems [75].

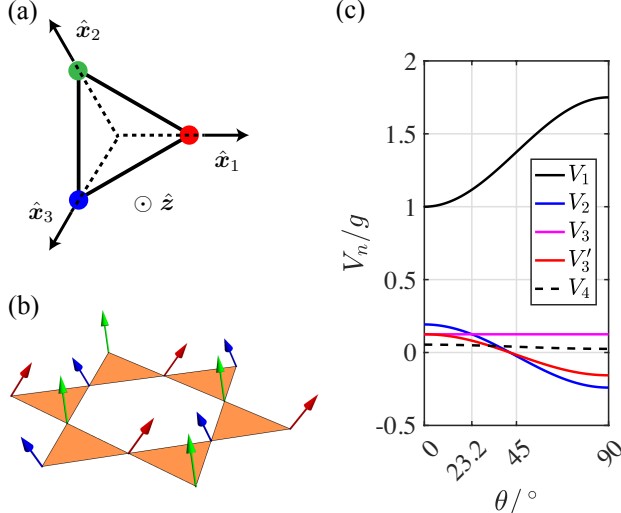

FIG. 7. (a) The local $x$-axis $\boldsymbol{x}_i$ and the global $z$-axis. The local axis $\hat{\boldsymbol{z}}_i = \hat{\boldsymbol{z}}\cos\theta + \hat{\boldsymbol{x}}\sin\theta$. (b) A generic non-collinear magnetic order. Red, green, and blue dots and arrows refer to the three different kagome sublattices. (c) The strength of the $n$th neighbor interaction $V_n$ as a function of $\theta$.

### B. Material's relevance and polar molecules

In this part, we discuss the material's relevance and the polar molecules. We start with the kagome dipolar magnets. One candidate family of materials is the rare-earth-based tripod magnets, $RE_3Mg_2Sb_3O_{14}$ and $RE_3Zn_2Sb_3O_{14}$ [29, 30, 100]. So far, owing to the connection with the pyrochlore spin ice physics, the existing discussion is based on the kagome version of the classical spin ice, and the long-range dipole-dipole interaction can be incorporated into the discussion of the kagome spin ice [77, 101–105]. It is certainly true that the kagome spin ice physics should be applicable to a few members of these tripod kagome materials, especially in the finite temperature regime where the thermal fluctuations govern the physics. One important difference between the tripod kagome system from the pyrochlore spin ice is that the local $\hat{\boldsymbol{z}}_i$ axis for the tripod kagome system no longer points to the center of the tetrahedron [30]. The local $\hat{\boldsymbol{z}}_i$ axis is modified by the non-magnetic ions ($Mg^{2+}$ and $Zn^{2+}$) above and below the kagome layer. We introduce an angular variable $\theta$ to parametrize the deviation of the local $\hat{\boldsymbol{z}}_i$ from the normal direction ($\hat{z}$) of the kagome plane (see Fig. 7), and depict the dependence of the $V_{ij}$'s on $\theta$.

To be concrete, the local $\hat{\boldsymbol{z}}_i$ is given by $\hat{\boldsymbol{z}}_i = \hat{\boldsymbol{z}}\cos\theta + \hat{\boldsymbol{x}}_i\sin\theta$, where $\hat{\boldsymbol{x}}_i$'s are defined on three sublattices and point outwards from the center of the triangle (see Fig. 7). The non-collinear local $\hat{\boldsymbol{z}}_i$'s make the dipole moments at different sites mismatch, modifying the relative strength of the $n$th neighbor interaction $V_n$ (see Fig. 7). It turns out that a finite $\theta$ could actually drive the dipolar interactions closer to the BFG model, helping the stabilization of $\mathbb{Z}_2$ spin liquid. For example, if $\theta \approx 23.2°$,

then the nonuniformity between $V_2$ and $V_3$ is completely eliminated, while the unwanted $V_3'$ interaction is weakened. Therefore, the non-collinear local axes in real tripod kagome materials add more degrees of freedom that can help stabilize $\mathbb{Z}_2$ spin liquids. The angle $\theta$ can be controlled by ambient pressure or chemical pressure. Ideally, it is likely to see the quantum phase transition from mundane phases to spin liquids by varying the pressure.

For $RE_3BWO_9$ and $Ca_3RE_3(BO_3)_5$, there are extra complications [106–108]. In addition to the different local axes, the kagome lattice is actually distorted from the perfect one. This requires more analysis of the actual dipole-dipole couplings under these lattice distortions. Moreover, there are the interlayer couplings. If the interlayer couplings are weak and do not destabilize the $\mathbb{Z}_2$ topological order in the 2D limit, our results can be applied. When the interlayer coupling is strong, there are two possibilities. In one possibility, the $\mathbb{Z}_2$ topological order is destroyed and a trivial state is obtained. In the other possibilities, the frustrated and long-range 3D interactions may stabilize the $\mathbb{Z}_2$ topological order in 3D.

For the realization in the polar molecular systems, the commonly used systems are $^{40}K^{87}Rb$ and $^{23}Na^{40}K$ molecules [33–36]. These molecules can be trapped on a kagome optical lattice. Since these molecules break the parity, a dipole moment is carried by these molecules, which naturally yields a dipolar interaction between these dipoles. These dipoles can be coupled to an external electric field. It is possible to design the profile of the electric field to tune the direction of the local $\hat{\boldsymbol{z}}$ axis, by which one can realize the equivalent magnetic orders as the tripod kagome materials.

To summarize, we have shown the possibility of search-

ing spin liquids in the dipolar kagome systems. These systems can be described by Ising spins with long-range dipolar interactions and other quantum interactions. The interactions endow the model with a similar structure as the Balents-Fisher-Girvin model. The elementary excitations, spinons, and visons can be described by an emergent $\mathbb{Z}_2$ lattice gauge theory and its dual theory. Meanfield theory shows the stability of fractionalization. The spectroscopic and thermodynamic experiments can measure the properties of these emergent and fractionalized excitations.

A direction that can be explored in future studies is the numerical demonstration of the $\mathbb{Z}_2$ spin liquids in the kagome dipolar systems. Equipped with the density-matrix renormalization group (DMRG) [109, 110], the ground states and then the static spin structure factors can be computed. Ideally, a phase diagram can be worked out. If $\mathbb{Z}_2$ spin liquids do exist in the phase diagram, one can calculate the dynamic spin structure factor using tDMRG [111], and examine the fractionalized excitations. Due to the long-range and anisotropy of the dipolar interaction, however, the bond dimension of the matrix product states must be sufficiently large, and the system size cannot be too small. Finally, it was shown that the 2D topological order is not variationally robust [112]. If the phase region for the topological order is quite narrow, it is likely that the system may be trapped in a metastable state whose energy differs from the topological ordered ground state by a sub-extensive amount. Small system sizes may not be able to resolve these competing states well. Owing to these challenges, we leave the DMRG studies of dipolar systems to a separate work.

## ACKNOWLEDGMENTS

We thank Bowen Ma, Si-Yu Pan, Zhaoming Tian, Martin Mourigal, Marcus Bintz, and Bo Yan for the fruitful discussions. This work is supported by the Ministry of Science and Technology of China with Grants No. 2021YFA1400300, and by the Fundamental Research Funds for the Central Universities, Peking University.

## Appendix A: Lattice structures

The lattice constant of the kagome lattice is set to 1. The centers of the hexagonal plaquettes in the kagome lattice form a triangular lattice. The dual lattice of the triangular lattice is a honeycomb lattice. The lattice constant of the triangular lattice is 2 and of the honeycomb lattice is $2/\sqrt{3}$. The triangular lattice and the honeycomb lattice have the same lattice vectors

$$\boldsymbol{a}_1 = (\sqrt{3}, -1), \quad \boldsymbol{a}_2 = (\sqrt{3}, 1). \quad \text{(A1)}$$

The reciprocal lattice vectors are

$$\boldsymbol{b}_1 = \pi(\frac{1}{\sqrt{3}}, -1), \quad \boldsymbol{b}_2 = \pi(\frac{1}{\sqrt{3}}, 1). \quad \text{(A2)}$$

For the honeycomb lattice with $\pi$ flux, the unit cell is enlarged to the magnetic unit cell, whose lattice vectors are

$$\boldsymbol{a}_1' = (\sqrt{3}, 1), \quad \boldsymbol{a}_2' = (0, 4). \quad \text{(A3)}$$

The reciprocal lattice vectors are

$$\boldsymbol{b}_1' = \pi(\frac{2}{\sqrt{3}}, 0), \quad \boldsymbol{b}_2' = \pi(-\frac{1}{2\sqrt{3}}, \frac{1}{2}). \quad \text{(A4)}$$

We give the definitions of high-symmetry points that were used in the main text

$$\Gamma = \Gamma_{00} : 0, \quad \Gamma_{11} : \boldsymbol{b}_1 + \boldsymbol{b}_2,$$
$$\text{M} : \frac{1}{2}(\boldsymbol{b}_1 + \boldsymbol{b}_2), \quad \text{K} : \frac{1}{4}(2\boldsymbol{b}_1 + \boldsymbol{b}_2). \quad \text{(A5)}$$

## Appendix B: Diagonalization of the bosonic spinon BdG Hamiltonian

We present the details of the diagonalization of the following Hamiltonian

$$\mathcal{H} = h_x \mathcal{B} \sum_{\boldsymbol{R}\boldsymbol{R}'} (b_{\boldsymbol{r}}^\dagger + b_{\boldsymbol{r}})(b_{\boldsymbol{r}'}^\dagger + b_{\boldsymbol{r}'}) + \lambda \sum_{\boldsymbol{r}} b_{\boldsymbol{r}}^\dagger b_{\boldsymbol{r}}. \quad \text{(B1)}$$

The sum over nearest-neighbor links can be rewritten as

$$\mathcal{H} = \frac{h_x \mathcal{B}}{2} \sum_{\boldsymbol{r}} \sum_{\boldsymbol{\delta}} (b_{\boldsymbol{r}}^\dagger + b_{\boldsymbol{r}})(b_{\boldsymbol{r}+\boldsymbol{\delta}}^\dagger + b_{\boldsymbol{r}+\boldsymbol{\delta}}) + \lambda \sum_{\boldsymbol{r}} b_{\boldsymbol{r}}^\dagger b_{\boldsymbol{r}}, \quad \text{(B2)}$$

where $\boldsymbol{\delta}$ runs in the vectors that connect the nearest neighbor. For the triangular lattice, they are

$$\pm\boldsymbol{a}_1, \quad \pm\boldsymbol{a}_2, \quad \pm(-\boldsymbol{a}_1 + \boldsymbol{a}_2). \quad \text{(B3)}$$

Under the Fourier transformation

$$b_{\boldsymbol{r}} = \frac{1}{\sqrt{N_\text{T}}} \sum_{\boldsymbol{k}}^{\text{B.Z.}} b_{\boldsymbol{k}} e^{i\boldsymbol{k}\cdot\boldsymbol{r}}, \quad \text{(B4)}$$

$\mathcal{H}$ becomes

$$\mathcal{H} = \frac{h_x \mathcal{B}}{2} \sum_{\boldsymbol{k}}^{\text{B.Z.}} \begin{pmatrix} b_{\boldsymbol{k}}^\dagger & b_{-\boldsymbol{k}} \end{pmatrix} \begin{pmatrix} \gamma_{\boldsymbol{k}} + \frac{\lambda}{h_x \mathcal{B}} & \gamma_{\boldsymbol{k}} \\ \gamma_{\boldsymbol{k}} & \gamma_{\boldsymbol{k}} + \frac{\lambda}{h_x \mathcal{B}} \end{pmatrix} \begin{pmatrix} b_{\boldsymbol{k}} \\ b_{-\boldsymbol{k}}^\dagger \end{pmatrix}$$
$$- \frac{\lambda N_\text{T}}{2}, \quad \text{(B5)}$$

where

$$\gamma_{\boldsymbol{k}} = \sum_{\delta} e^{i\boldsymbol{k}\cdot\boldsymbol{\delta}} = 2\left[ 2\cos\left(\sqrt{3}k_x\right)\cos(k_y) + \cos(2k_y) \right] \quad \text{(B6)}$$

is the structure factor of the triangular lattice.

$\mathcal{H}$ is a bosonic BdG Hamiltonian. When the 2-by-2 matrix is positively definite, the Hamiltonian is stable and can be diagonalized by para-unitary matrices. The diagonal elements are given by the eigenvalues of the following matrix

$$\begin{pmatrix} \gamma_{\boldsymbol{k}} + \frac{\lambda}{h_x \mathcal{B}} & \gamma_{\boldsymbol{k}} \\ -\gamma_{\boldsymbol{k}} & -\gamma_{\boldsymbol{k}} + \frac{\lambda}{h_x \mathcal{B}}. \end{pmatrix} \quad \text{(B7)}$$

After the diagonalization, we obtain

$$\mathcal{H} = \frac{1}{2} \sum_{\boldsymbol{k}}^{\text{B.Z.}} \left( \omega_{\boldsymbol{k}} a_{\boldsymbol{k}}^\dagger a_{\boldsymbol{k}} - \omega_{\boldsymbol{k}} a_{-\boldsymbol{k}} a_{-\boldsymbol{k}}^\dagger \right) - \frac{\lambda N_{\text{T}}}{2}, \quad \text{(B8)}$$

where

$$\omega_{\boldsymbol{k}} = \sqrt{\lambda \left( \lambda + 2 h_x \mathcal{B} \gamma_{\boldsymbol{k}} \right)}. \quad \text{(B9)}$$

Reorganizing the bosonic operators, we obtain

$$\mathcal{H} = \sum_{\boldsymbol{k}}^{\text{B.Z.}} \left( \omega_{\boldsymbol{k}} a_{\boldsymbol{k}}^\dagger a_{\boldsymbol{k}} + \frac{1}{2} \right) - \frac{\lambda N_{\text{T}}}{2}. \quad \text{(B10)}$$

**Appendix C: Diagonalization of even $\mathbb{Z}_2$ vison Hamiltonian**

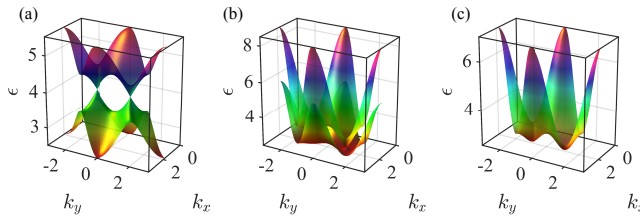

FIG. A1. Vison dispersion on the honeycomb lattice with zero flux. (a) $J_1 = 1$ and $J_2 = 0$; (b) $J_1 = 1$ and $J_2 = 1$; (c) $J_1 = 0$ and $J_2 = 1$. Chemical potential is set to $\mu = 4$.

We present the details of the diagonalization of the following Ising Hamiltonian

$$\mathcal{H} = J_1 \sum_{\boldsymbol{R}\boldsymbol{R}'} \mu_{\boldsymbol{R}}^z \mu_{\boldsymbol{R}'}^z + J_2 \sum_{\boldsymbol{R}\boldsymbol{R}'} \mu_{\boldsymbol{R}}^z \mu_{\boldsymbol{R}'}^z. \quad \text{(C1)}$$

In principle, we can use the hard-core boson representation of vison creation operator $\mu^z$ and diagonalize a bosonic BdG Hamiltonian to obtain the vison dispersion, as we have done for spinons. This process is actually equivalent to solving the soft modes of Ising model. To proceed, we define the Fourier transformation,

$$\mu_{\boldsymbol{R}}^z = \frac{1}{\sqrt{N_{\text{H}}}} \sum_{\boldsymbol{k}}^{\text{B.Z.}} \mu_{\boldsymbol{k}}^z e^{i\boldsymbol{k}\cdot\boldsymbol{R}}. \quad \text{(C2)}$$

Under the transformation, $\mu_{\boldsymbol{k}}^z$ is no longer an Ising operator because of the constraint $(\mu_{\boldsymbol{R}}^z)^2 = 1$. We can

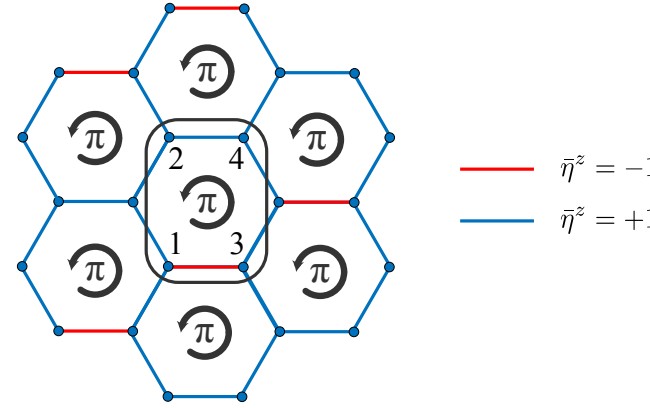

FIG. A2. The honeycomb lattice with $\pi$ flux in each hexagon. The gauge field on blue links is $\bar{\eta}^z = +1$ and in red links is $\bar{\eta}^z = -1$. The magnetic unit cell contains four sites.

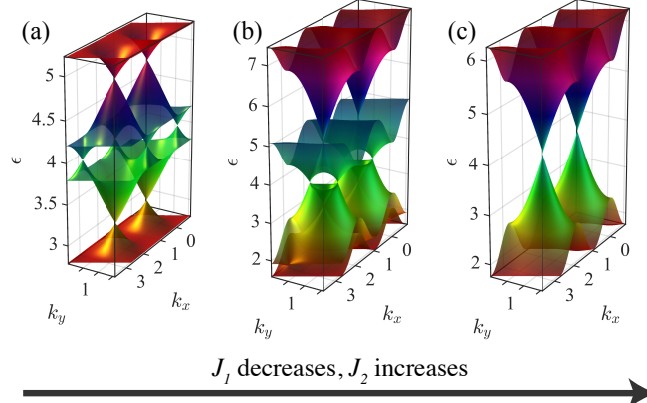

$J_1$ decreases, $J_2$ increases

FIG. A3. Four energy bands of visons on the honeycomb lattice with $\pi$ flux. (a) $J_1 = 1$ and $J_2 = 0$; (b) $J_1 = 1$ and $J_2 = 1$; (c) $J_1 = 0$ and $J_2 = 1$.

remove the constraint by adding a mean-field chemical potential term $\mu \sum_{\boldsymbol{R}} [(\mu_{\boldsymbol{R}}^z)^2 - 1]$. Finally, we obtain

$$\mathcal{H} = \sum_{\boldsymbol{k}}^{\text{B.Z.}} \left( \mu_{1,-\boldsymbol{k}}^z \ \mu_{2,-\boldsymbol{k}}^z \right) \begin{pmatrix} \mu + \frac{J_2}{2}\gamma_{\boldsymbol{k}} & \frac{J_1}{2}\zeta_{\boldsymbol{k}} \\ \frac{J_1}{2}\zeta_{\boldsymbol{k}}^* & \mu + \frac{J_2}{2}\gamma_{\boldsymbol{k}} \end{pmatrix} \begin{pmatrix} \mu_{1,\boldsymbol{k}}^z \\ \mu_{2,\boldsymbol{k}}^z \end{pmatrix}, \quad \text{(C3)}$$

where

$$\gamma_{\boldsymbol{k}} = \sum_{\delta} e^{i\boldsymbol{k}\cdot\boldsymbol{\delta}} = 2 \left[ 2 \cos\left(\sqrt{3}k_x\right) \cos(k_y) + \cos(2k_y) \right] \quad \text{(C4)}$$

is the structure factor of the triangular lattice, and

$$\zeta_{\boldsymbol{k}} = 1 + e^{i\boldsymbol{k}\cdot\boldsymbol{a}_1} + e^{i\boldsymbol{k}\cdot\boldsymbol{a}_2} = 1 + 2e^{-i\sqrt{3}k_x} \cos(k_y) \quad \text{(C5)}$$

is the structure factor of the honeycomb lattice. Diagonalizing the 2-by-2 matrix, we obtain

$$\mathcal{H} = \sum_{\boldsymbol{k}}^{\text{B.Z.}} \left( \epsilon_{-,\boldsymbol{k}} \mu_{-,-\boldsymbol{k}}^z \mu_{-,\boldsymbol{k}}^z + \epsilon_{+,\boldsymbol{k}} \mu_{+,-\boldsymbol{k}}^z \mu_{+,\boldsymbol{k}}^z \right), \quad \text{(C6)}$$

with energy bands

$$\epsilon_{\pm,\boldsymbol{k}} = \mu + \frac{J_2}{2}\gamma_{\boldsymbol{k}} \pm \frac{|J_1|}{2}|\zeta_{\boldsymbol{k}}|. \tag{C7}$$

When tuning the relative strength of $J_1$ and $J_2$, one would see a crossover from the honeycomb lattice dispersion to the triangular lattice dispersion. In Fig. A1, we show three cases: (a) $J_1 = 1$ and $J_2 = 0$, the dispersion is dominated by the honeycomb lattice part; (b) $J_1 = 1$ and $J_2 = 1$, the dispersion is a mix of the honeycomb lattice part and the triangular lattice part; (c) $J_1 = 0$ and $J_2 = 1$, the dispersion is purely from the triangular lattice part.

## Appendix D: Diagonalization of odd $\mathbb{Z}_2$ vison Hamiltonian

We present the details of the diagonalization of the following Hamiltonian

$$\mathcal{H} = J_1 \sum_{\boldsymbol{R}\boldsymbol{R}'} \bar{\eta}^z_{\boldsymbol{R}\boldsymbol{R}'}\mu^z_{\boldsymbol{R}}\mu^z_{\boldsymbol{R}'} + J_2 \sum_{\boldsymbol{R}\boldsymbol{R}'} \bar{\eta}^z_{\boldsymbol{R}\boldsymbol{R}''}\bar{\eta}^z_{\boldsymbol{R}''\boldsymbol{R}'}\mu^z_{\boldsymbol{R}}\mu^z_{\boldsymbol{R}'}, \tag{D1}$$

where the gauge field $\bar{\eta}^z_{\boldsymbol{R}\boldsymbol{R}'} = 1$ for the blue links and $-1$ for the red links (see Fig. A2). Such a gauge choice leads to a uniform $\pi$ flux pattern. The unit cell is enlarged to the magnetic unit cell, which contains four sites.

To study the vison dispersion, we calculate the soft modes of the Ising Hamiltonian $\mathcal{H}$. After the Fourier transformation Eq. (C2), we obtain

$$\begin{aligned}
\mathcal{H} &= \frac{J_1}{2} \sum_{\boldsymbol{k}}^{\text{B.Z.}} \begin{pmatrix} \mu^z_{1,-\boldsymbol{k}} & \mu^z_{2,-\boldsymbol{k}} & \mu^z_{3,-\boldsymbol{k}} & \mu^z_{4,-\boldsymbol{k}} \end{pmatrix} \begin{pmatrix} & & -1+e^{i\boldsymbol{k}\cdot\boldsymbol{a}'_1} & e^{i\boldsymbol{k}\cdot\boldsymbol{a}'_1} \\ & & e^{i\boldsymbol{k}\cdot(\boldsymbol{a}'_1-\boldsymbol{a}'_2)} & 1+e^{i\boldsymbol{k}\cdot\boldsymbol{a}'_1} \\ -1+e^{-i\boldsymbol{k}\cdot\boldsymbol{a}'_1} & e^{-i\boldsymbol{k}\cdot(\boldsymbol{a}'_1-\boldsymbol{a}'_2)} & & \\ e^{-i\boldsymbol{k}\cdot\boldsymbol{a}'_1} & 1+e^{-i\boldsymbol{k}\cdot\boldsymbol{a}'_1} & & \end{pmatrix} \begin{pmatrix} \mu^z_{1,\boldsymbol{k}} \\ \mu^z_{2,\boldsymbol{k}} \\ \mu^z_{3,\boldsymbol{k}} \\ \mu^z_{4,\boldsymbol{k}} \end{pmatrix} \\
&+ \frac{J_2}{2} \sum_{\boldsymbol{k}}^{\text{B.Z.}} \begin{pmatrix} \mu^z_{1,-\boldsymbol{k}} & \mu^z_{2,-\boldsymbol{k}} & \mu^z_{3,-\boldsymbol{k}} & \mu^z_{4,-\boldsymbol{k}} \end{pmatrix} \begin{pmatrix} -2\cos(\boldsymbol{k}\cdot\boldsymbol{a}'_1) & \alpha_{\boldsymbol{k}} & & \\ \alpha^*_{\boldsymbol{k}} & 2\cos(\boldsymbol{k}\cdot\boldsymbol{a}'_1) & & \\ & & -2\cos(\boldsymbol{k}\cdot\boldsymbol{a}'_1) & \beta_{\boldsymbol{k}} \\ & & \beta^*_{\boldsymbol{k}} & 2\cos(\boldsymbol{k}\cdot\boldsymbol{a}'_1) \end{pmatrix} \begin{pmatrix} \mu^z_{1,\boldsymbol{k}} \\ \mu^z_{2,\boldsymbol{k}} \\ \mu^z_{3,\boldsymbol{k}} \\ \mu^z_{4,\boldsymbol{k}} \end{pmatrix} \\
&+ \mu \sum_{\boldsymbol{k}}^{\text{B.Z.}} \begin{pmatrix} \mu^z_{1,-\boldsymbol{k}} & \mu^z_{2,-\boldsymbol{k}} & \mu^z_{3,-\boldsymbol{k}} & \mu^z_{4,-\boldsymbol{k}} \end{pmatrix} \begin{pmatrix} 1 & & & \\ & 1 & & \\ & & 1 & \\ & & & 1 \end{pmatrix} \begin{pmatrix} \mu^z_{1,\boldsymbol{k}} \\ \mu^z_{2,\boldsymbol{k}} \\ \mu^z_{3,\boldsymbol{k}} \\ \mu^z_{4,\boldsymbol{k}} \end{pmatrix}
\end{aligned} \tag{D2}$$

$$\alpha_{\boldsymbol{k}} = 1 + e^{i\boldsymbol{k}\cdot\boldsymbol{a}'_2} + e^{i\boldsymbol{k}\cdot\boldsymbol{a}'_1} - e^{i\boldsymbol{k}\cdot(\boldsymbol{a}'_2-\boldsymbol{a}'_1)}, \quad \beta_{\boldsymbol{k}} = 1 + e^{i\boldsymbol{k}\cdot\boldsymbol{a}_2} - e^{i\boldsymbol{k}\cdot\boldsymbol{a}'_1} + e^{i\boldsymbol{k}\cdot(\boldsymbol{a}'_2-\boldsymbol{a}'_1)}. \tag{D3}$$

The analytical diagonalization of the 4-by-4 matrix is impossible. Thus, we obtain the vison dispersion by numerical diagonalization. In Fig. A3, we plot the vison dispersion for three cases: (a) $J_1 = 1$ and $J_2 = 0$, the dispersion is dominated by the honeycomb lattice part; (b) $J_1 = 1$ and $J_2 = 1$, the dispersion is a mix of the honeycomb lattice part and the triangular lattice part; (c) $J_1 = 0$ and $J_2 = 1$, the dispersion is purely from the triangular lattice part.

## Appendix E: The effects of $J^\perp$ spin-flipping terms

The formation of spin liquids significantly relies on the quantum dynamics of the ground state manifold. In the main text, we have focused on the quantum dynamics generated by the transverse field $h_x$. Actually, there exist perpendicular $J^\perp_{ij}$ spin-flipping terms in kagome dipolar magnets, which can provide similar quantum dynamics as the transverse field $h_x$ but have some quantitative differences. Thus, we here address some key aspects of $J^\perp_{ij}$ spin-flipping terms.

To be concrete, we rewrite the kagome dipolar Hamiltonian Eq. (2) with the nearest neighbor $J^\perp_1$ spin-flipping terms explicitly,

$$\begin{aligned}
H &= \frac{1}{2} \sum_{i\neq j} V_{ij} S^z_i S^z_j + \sum_{\langle ij \rangle} J^\perp_1 (S^x_i S^x_j + S^y_i S^y_j) \\
&\quad - h_z \sum_i S^z_i - h_x \sum_i S^x_i + \cdots .
\end{aligned} \tag{E1}$$

What results would change in the main text when we include $J^\perp_1$ terms? From the perturbative point of view in Sec. IV, the dimer-flipping dynamics is generated on the BFG or restricted BFG ground state manifold at the second-order perturbation of $J^\perp_1$, which is generally stronger than that of the transverse field $h_x$.

From the non-perturbative mean-field point of view in Sec. V, $J^\perp_1$ terms lead to higher-order hoppings of

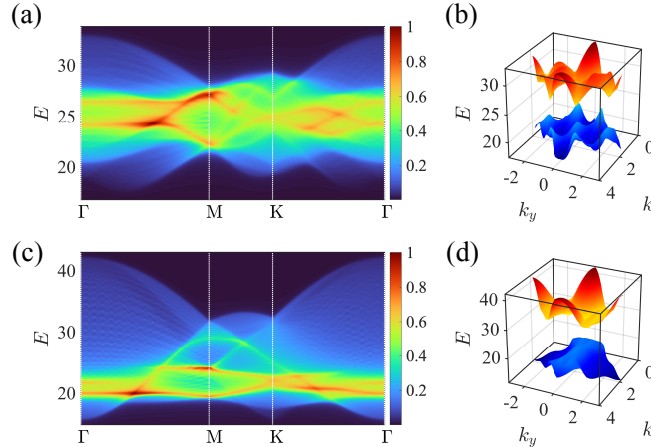

(a) (b) (c) (d)

FIG. A4. The spinon continuum for $J_1^\perp = 1$ and $h_x = 1$. (a) and (b) are the case of $\mathcal{B} > 0$, the parameters are $\mathcal{V} = 30$, $h_z = -15$, $\lambda = 12.53$ and $\mathcal{B} = 0.5769$; (c) and (d) are the case of $\mathcal{B} > 0$, the parameters are $\mathcal{V} = 30$, $h_z = -15$, $\lambda = 12.54$ and $\mathcal{B} = -0.5768$. (a) and (c) are the spinon continuum along high-symmetry lines; (b) and (d) show the lower and upper limits of the continuum

spinons. To see this, we recall the mapping

$$S_i^x \to \frac{1}{2}\sigma_{\boldsymbol{rr'}}^z, \quad S_i^y \to -\frac{1}{2}\sigma_{\boldsymbol{rr'}}^y, \quad S_i^z \to \frac{1}{2}\sigma_{\boldsymbol{rr'}}^x. \quad \text{(E2)}$$

Under the gauge constraint Eq. (20), $\sigma_{\boldsymbol{rr'}}^z$ is mapped to $\sigma_{\boldsymbol{r}}^z\tau_{\boldsymbol{r}}^z\tau_{\boldsymbol{r'}}^z$ and $\sigma_{\boldsymbol{rr'}}^y$ is mapped to $\sigma_{\boldsymbol{r}}^y\tau_{\boldsymbol{r}}^y\tau_{\boldsymbol{r'}}^y$. Then, after choosing the mean-field ansatz that condenses the gauge field, $\langle\sigma_i^x\rangle = 1 - \mathcal{B}^2$, $\langle\sigma_i^y\rangle = 0$, and $\langle\sigma_i^z\rangle = 2\mathcal{B}$, one can see that the $J_1^\perp$ terms become

$$\sum_{\langle ij \rangle} J_1^\perp (S_i^x S_j^x + S_i^y S_j^y) \sim 2J_1^\perp \mathcal{B}^2 \sum_{\boldsymbol{r_1 r_2}} \sum_{\boldsymbol{r_3} \neq \boldsymbol{r_1}} \tau_{\boldsymbol{r_1}}^z \tau_{\boldsymbol{r_3}}^z, \quad \text{(E3)}$$

where $\boldsymbol{r_1 r_2}$ and $\boldsymbol{r_2 r_3}$ are triangular lattice links. These terms are reduced to the nearest-neighbor and next-nearest-neighbor hoppings of spinon. Considering the next-nearest-neighbor hoppings, the spinon energy band now becomes $\omega_{\boldsymbol{k}} = \sqrt{\lambda^2 - 2h_x\mathcal{B}\lambda\gamma_{\boldsymbol{k}} + 4J_1^\perp\mathcal{B}^2\lambda(\gamma_{\boldsymbol{k}} + \gamma'_{\boldsymbol{k}})}$, where $\gamma'_{\boldsymbol{k}} = 2\left[2\cos\left(\sqrt{3}k_x\right)\cos(3k_y) + \cos(2\sqrt{3}k_y)\right]$ is the structure factor of the next-nearest-neighbor links of the triangular lattice. In the presence of $J_1^\perp$, the four kinds of $\mathbb{Z}_2$ spin liquids still exist in the mean-field phase diagram, the only difference is a quantitative shift of the phase boundaries. Moreover, as $J_1^\perp$ changes the spinon energy band, the spinon continuum also changes. The case of $\mathcal{B} > 0$ and $\mathcal{B} < 0$ is no longer symmetric (see Fig. A4).

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
