# Peer review of "Z2 topological orders in kagome dipolar systems: Feedback from Rydberg quantum simulator"

_SciPost Physics_

## Round 1 · Referee Report · Anonymous (Referee 1) · 2025-5-22

Strengths
1- Timely and exciting topic. 2- Detailed study of physically relevant model systems . 3 - Comprehensive theoretical approach including analogies to well established models using complementary perturbative approaches. 4- detailed characterization of spinon and vison excitations, including their continuum spectra and how these spectra differ for various types of Z2 liquids. 5- Novel insights into the effect of dipolar interactions on spin liquids.
Weaknesses
1- The mean filed and perturbative approaches do have limitations. The authors acknowledge in the outlook the need for numerical studies like DMRG to confirm the phase diagram and dynamics, especially given the long-range and anisotropic nature of dipolar interactions. 2- Several other, potentially symmetry breaking, phases might occur instead of the sought after quantum spin liquid. It might be useful to make the reader more aware of the assumptions made. 3- The large number of parameters used in the models lead naturally to a huge phase space.
Report
The manuscript presents a significant and timely theoretical contribution to the search for exotic quantum phases, specifically Z2 topological orders, in kagome dipolar systems. The comprehensive theoretical treatment including concrete predictions for experimental signatures, makes this work a valuable resource for both theorists and experimentalists in the field of frustrated magnetism and topological quantum matter.
There are a number of points the author should consider before I can recommend the manuscript for publication.
(1) The study relies on mean-field and perturbative methods, which possess inherent limitations. While the focus is on the quantum spin liquid state, the potential for competing phases, including those involving symmetry-breaking, should be considered. A more explicit discussion of the assumptions favoring the quantum spin liquid phase could be helpful.
(2) The authors might want to add a discussion mention existing works predicting dynamical signatures of quantum spin liquids (e.g., Phys. Rev. Lett. 132, 066702, Phys. Rev. X 8, 011037 , Phys. Rev. Lett. 118, 227201, Phys. Rev. Lett. 124, 097204 , ...).
(3) The are still several typos through the manuscript. Moreover, I think that the commonly used convention is "kagome" (without accented "é").
Requested changes
1- Make it more clear what the underlying assumption are and what the potential computing phases are 2- Compare to the existing literature on dynamical probes.
Recommendation
Ask for minor revision
Summary of Changes
At the end of Sec. V B, a discussion about possible competing phases is added.
In the beginning of the 4th paragraph of Sec. I, “Since the Z2 topological order has not yet been fully discovered in any existing experiment” is revised to “While Z2 topological order has appeared in Rydberg atom experiments and quantum simulators, its evidence in real condensed matter materials has not yet been fully discovered in any existing experiment. Therefore, it is more important for us to address the key experimental features that distinguish it from other states.”
At the end of Sec. VII A 1, a discussion about measuring dynamical signatures of fractional excitations is added.
“kagomé” is replaced by “kagome”.
At the end of Sec. IV A, “For mz = msat/3$, recent numerical simulations suggest that the Z2 liquid phase also lives in the finite μ region, and a hidden order is reported near and above μ=0” is added.
Several typos are fixed.
Response to Referee 1
We thank Referee 1 for detailed comments and thoroughly reading of our manuscript. Below, we list our responses point-by-point to the comments raised. Referee’s comments are highlighted by gray shade.
Referee 1:
The study relies on mean-field and perturbative methods, which possess inherent limitations. While the focus is on the quantum spin liquid state, the potential for competing phases, including those involving symmetry-breaking, should be considered. A more explicit discussion of the assumptions favoring the quantum spin liquid phase could be helpful.
Our response:
We thank Referee 1 for bringing up this important question. Competition between the mean-field Ansatz and other potential phases is indeed a fundamental question of mean-field approximation. In the kagome Van der Waals Rydberg atom system, as studied in Ref. [24] [R. Samajdar, et. al., PNAS 118, e2015785118 (2021)], there are symmetry-breaking ordered phases such as the paramagnetic phase, staggered phase, nematic phase, stripe phase, and string phase that competes with the Z_2 spin liquid phase. Considering the similar interaction structure, we expect these order phases could also appear in our kagome dipolar systems, although their energy and location in the phase diagram could be different. This can be confirmed by numerical scan of phase diagram using methods such as DMRG, which we leave for feature study. In our mean-field calculation, the assumption we made is that the ground state is a string-net-condensed state with condensed gauge field, such that the matter field (spinons) is deconfined. The liquid phase is protected by a finite spinon gap. As long as the gap is finite, spinons will not condense, which stabilizes the Z_2 spin liquid phase. We also add a discussion about the underlying assumptions in the revised version (see Change 1).
Referee 1:
The authors might want to add a discussion mention existing works predicting dynamical signatures of quantum spin liquids (e.g., Phys. Rev. Lett. 132, 066702, Phys. Rev. X 8, 011037 , Phys. Rev. Lett. 118, 227201, Phys. Rev. Lett. 124, 097204 , ...).
Our response:
Thanks for reminding us these existing works. We have added a discussion about the detection of fractional excitations using threshold spectroscopic experiments (see Change 3).
Referee 1:
The are still several typos through the manuscript. Moreover, I think that the commonly used convention is "kagome" (without accented "é").
Our response:
We apologize for these typos. In revised version, we have fixed most of these typos we found. We have also replaced “kagomé” by “kagome” accordingly (see Change 4).

Author: Gang Chen on 2025-06-11 [id 5560]
(in reply to Report 2 on 2025-05-23)Response to Referee 2
We thank Referee 2 for detailed comments and thoroughly reading of our manuscript. Below, we list our responses point-by-point to the comments raised. Referee’s comments are highlighted by gray shade.
Referee 2:
In the Introduction, the authors write that “the Z2 topological order has not yet been fully discovered in any existing experiment’’—this is misleading. Experiments with Rydberg atom arrays [Semeghini et al., Science (2021)] as well as with superconducting circuits by Google Quantum AI [Satzinger et al., Science (2021)] have indeed presented convincing signatures of a Z2 spin liquid, or equivalently, toric-code topological order.
Our response:
We appreciate Referee 2 for pointing out this inaccurate expression. Accordingly, we have revised this sentence as “While Z_2 topological order has appeared in Rydberg atom experiments and quantum simulators, its evidence in real condensed matter materials has not yet been fully discovered in any existing experiment.” (see Change 2).
Referee 2:
In Sec. II, the authors claim that “the superexchange between the transverse spin components could be relatively weak.’’ For a candidate material like Ho_3 Mg_2 Sb_3 O_14, how does J_ij^⊥ compare to V_ij? Please provide actual numbers.
Our response:
As shown in Ref. [30] [Z. Dun, et. al., Physical Review X 10, 031069 (2020)], Ho_3 Mg_2 Sb_3 O_14 has a strong Ising anisotropy, whose superexchange between transverse spin components J_ij^⊥ should be much smaller compared with superexchange between Ising spin components J_ij^z. For nearest neighbors, J_ij^z≈-0.67K, which is already very weak compared with the dipole-dipole interaction D=1.69K between nearest neighbors. Thus, |J_ij^⊥|≪|J_ij^z |<|D|, it is reasonable to think J_ij^⊥ as relatively weak, while the actual value may depend on the material realization and should be measured by experiments.
Referee 2:
On page 4, the authors write that “the superexchange interaction of the 4f electrons is quite short-ranged and is often dominated by the first exchange’’, and this fact proves to be important for their analysis in the ferromagnetic case. However, the paper does not provide any references or evidence to support this claim.
Our response:
Superexchange originates from the electron hoppings between magnetic positive ions and the nonmagnetic negative ions. Electrons in 4f orbitals of the magnetic positive ions are highly localized for the most of rare-earth elements, which has been discussed in references such as [H. Kusunose, J. Phys. Soc. Jpn. 77, 064710 (2008), W. Witczak-Krempa, et. al., Annual Review of Condensed Matter Physics 5, 57 (2014).]. Therefore, the hopping strength t of 4f electrons among lattice sites is small. As the superexchange between two sites is proportional to t^n/U^(n-1), where U is the onsite repulsion and n is the number of hoppings needed to connect the two sites, the superexchange decay fast over distance. Therefore, we ignore higher-order superexchange and keep only the nearest-neighbor one. Accordingly, we’ve added these relevant references in the newer version of our manuscript.
Referee 2:
Below Eq. (10), the paper states that “μ=0 in the cases here.’’ Why? μ represents the interaction between parallel dimers, which corresponds to the interaction S_i^z S_j^z between two spins separated by ‖i-j‖. Since V_ij is long-ranged, this interaction is nonzero and thus μ≠0.
Our response:
In the derivation of Eq. (10), we have truncated the long-range dipole-dipole interaction at the 3rd nearest neighbors and ignored the inter-hexagon 3rd nearest neighbors V_3^'. Under this approximation, the effective Hamiltonian induced by spin-flipping terms on the BFG ground states manifold is simply a bow-tie ring exchange. The effective Hamiltonian is identical to the dimer-flipping term in the Eq. (10), while the 2nd row of Eq. (10) is absent as long as we are sticked to the approximation. Indeed, μ represents the interaction between parallel dimers, which could come from V_3^' interactions, as discussed in the last paragraph of Sec. IV A.
Referee 2:
In Sec. IV A, the authors mention that for “mz=msat/3, the liquid phase remains extended even for μ=0.’’ This is incorrect. For the case with one dimer per site, Fig. 3 of Ref. 43 clearly shows that the Z2 spin liquid is stable only for 0.7<μ/t≤1. For the case with two dimers per site, the most recent QMC calculations suggest that the Z2 spin liquid occurs for 0.59<μ/t≤1 [Plat et al., Phys. Rev. B 92, 174402 (2015); Ran et al., Commun. Phys. 7, 207 (2024)].
Our response:
We appreciate Referee 2 for pointing out this mistake. Ref. [43] cited in our manuscript claims that the liquid phase exists for -0.3<μ<1.0. However, as Referee 2’s reminder, in these newly published works, the result obtained by Ref. [43] seems to be incorrect. Therefore, we revised this sentence and added some discussions about this point (see Change 5).
Referee 2:
On page 6, “the 2nd row of Eq. (10)’’ should refer to the first row instead.
Our response:
Thanks for pointing out this typo. We have revised it accordingly (see Change 6).
Referee 2:
In deriving the mean-field theory, the authors say that “To determine the dispersion of the b bosons, we condense the B bosons’’. Why is condensing the B bosons a valid approximation inside the spin liquid phase?
Our response:
In the language of lattice gauge field theory, the spin liquid phase is interpreted as a string-net condensed state, in which gauge field is condensed. The condensation of the gauge field means the ground state contains all possible configurations of gauge field. As B bosons are bosons of the gauge field, the spin liquid phase naturally corresponds to the condensed state of B bosons.
Referee 2:
Sec. VI A asserts that “Assuming a uniform gauge field condensation, 〈S_i^x (t) S_i^x (0)〉 is proportional to the density of states of spinon continuum ρ_spinon (k,E) up to a nonuniversal form factor’’. This statement is then used to compute the spectral signatures of the spinon continuum, which is one of the main results of the paper. — However, in order to match the theory with any dynamical structure factor that one might measure experimentally, it is important to keep track of these form factors carefully. So, in order to take these plots (and also the ones for the vison continuum) seriously, please include a detailed derivation of these form factors on the kagome lattice in the revised manuscript. — To obtain the smooth continuum features shown in Fig. 4 from the discrete dispersion relations, I assume that some broadening has been employed. Please elaborate.
Our response:
We thank Referee 2 for raising this potential improvement of our work. The form factor indeed changes the detailed shape of the spinon continuum and vison continuum. However, form factor is a nonuniversal property of our model and it closely relies on the parameters. For different materials, the form factor could be different. Even if we track the form factor by equations, it is unlikely to obtain a quantitatively matching with experimental results. Instead, the existence of a continuum in the spectrum is universal and doesn’t depend on the materials, as long as the material is in a spin liquid phase, which is one of the most important pieces of information of this work. Therefore, we leave the detailed calculation to an ongoing quantitative work with a combination of the DMRG study.
As for the smooth continuum in Fig. 4, it is obtained by calculating the density of two-spinons states based on the momentum and energy conservation: k=q_1+q_2, E(k)=ω(q_1 )+ω(q_2). There isn’t any approximation such as broadening employed in this calculation.
Referee 2:
On page 8, it is written that the π “flux pattern in the triangular lattice doesn’t cause symmetry fractionalization but reverses the energy band.’’ Why?
Our response:
The π flux in triangular lattice with only nearest-neighbor hoppings can be realized by a gauge choice in which the gauge phase e^(iA_rr' )=-1 for all links. Since this gauge preserves all lattice symmetries of the triangular lattice, it doesn’t lead to symmetry fractionalization. All the effect of the π flux is to reverse the energy band up-side-down: ω(k)→-ω(k).
Referee 2:
In Sec. VII A.4, the authors write that “To access the spectroscopic properties of the Z2 liquid phase in the polar molecule systems, one can use the two-photon Raman spectroscopy [93] to obtain equivalent outcomes as the inelastic neutron scatterings for the spins.’’ This is a sweeping claim and has to be substantiated with details instead of deferring the responsibility to Ref. 93. In ultracold atomic systems, measurement of two-time correlation functions, as required for dynamical structure factors, is challenging because projective measurement collapses the wavefunction. Hence, it would be very useful if the authors could actually provide the schematic details for extracting such spectroscopic properties.
Our response:
Sorry, we are not experts in this specific measurement. Thus, we choose to cite the relevant works instead of delving into the details.
Referee 2:
Finally, in discussing the fate of RE3BWO9 and Ca3RE3(BO3) under the effect of kagome lattice distortions, the authors mention the possibility that “the frustrated and long-range 3D interactions may stabilize the Z2 topological order in 3D. ’’ Is there any known example of a model in which this scenario occurs?
Our response:
For example, it can be realized with quantum dimer model on FCC lattice or other frustrated 3D lattices, see (arxiv 0809.3051) for discussion. There are other models such as 3D toric code.

---

## Round 1 · Referee Report · Anonymous (Referee 2) · 2025-5-23

Strengths
- In the present manuscript, Zhao and Chen systematically investigate the possibility of topological order in kagome-lattice systems with dipolar interactions.
- They consider both solid-state realizations, in materials with Kramers doublets, non-Kramers doublets, and cluster Mott insulators, as well as newer synthetic systems such as ultracold polar molecules.
- Using a combination of mappings to quantum dimer models, perturbation theory, and a gauge-theoretic description, they argue for the existence of a $\mathbb{Z}_2$ spin liquid phase in various regimes and discuss the experimental signatures thereof.
- The work is timely and of experimental relevance to a number of systems.
Weaknesses
Before I can recommend publication of this work in SciPost Physics, there are several questions and comments that I would request the authors to address in a revised version first.
Report
-
In the Introduction, the authors write that “the $\mathbb{Z}_2$ topological order has not yet been fully discovered in any existing experiment’’—this is misleading. Experiments with Rydberg atom arrays [Semeghini et al., Science (2021)] as well as with superconducting circuits by Google Quantum AI [Satzinger et al., Science (2021)] have indeed presented convincing signatures of a $\mathbb{Z}_2$ spin liquid, or equivalently, toric-code topological order.
-
In Sec. II, the authors claim that “the superexchange between the transverse spin components could be relatively weak.’’ For a candidate material like Ho$_3$Mg$_2$Sb$_3$O$_{14}$, how does $J^\perp_{ij}$ compare to $V_{ij}$? Please provide actual numbers.
-
On page 4, the authors write that “the superexchange interaction of the 4$f$ electrons is quite short-ranged and is often dominated by the first exchange’’, and this fact proves to be important for their analysis in the ferromagnetic case. However, the paper does not provide any references or evidence to support this claim.
-
Below Eq. (10), the paper states that “$\mu =0$ in the cases here.’’ Why? $\mu$ represents the interaction between parallel dimers, which corresponds to the interaction $S^z_i S^z_j$ between two spins separated by $\vert \vert i -j \vert \vert$. Since $V_{ij}$ is long-ranged, this interaction is nonzero and thus $\mu \ne 0$.
-
In Sec. IV A, the authors mention that for “$m_z = m_{\mathrm{sat}}/3$, the liquid phase remains extended even for $\mu = 0$.’’ This is incorrect. For the case with one dimer per site, Fig. 3 of Ref. 43 clearly shows that the $\mathbb{Z}_2$ spin liquid is stable only for $0.7 < \mu/t \le 1$. For the case with two dimers per site, the most recent QMC calculations suggest that the $\mathbb{Z}_2$ spin liquid occurs for $0.59 < \mu/t \le 1$ [Plat et al., Phys. Rev. B 92, 174402 (2015); Ran et al., Commun. Phys. 7, 207 (2024)].
-
On page 6, “the 2nd row of Eq. (10)’’ should refer to the first row instead.
-
In deriving the mean-field theory, the authors say that “To determine the dispersion of the $b$ bosons, we condense the $B$ bosons’’. Why is condensing the $B$ bosons a valid approximation inside the spin liquid phase?
-
Sec. VI A asserts that “Assuming a uniform gauge field condensation, $\langle S^x_i (t) S^x_i (0)\rangle$ is proportional to the density of states of spinon continuum $\rho_{\textrm{spinon}}(k,E)$ up to a nonuniversal form factor’’. This statement is then used to compute the spectral signatures of the spinon continuum, which is one of the main results of the paper. — However, in order to match the theory with any dynamical structure factor that one might measure experimentally, it is important to keep track of these form factors carefully. So, in order to take these plots (and also the ones for the vison continuum) seriously, please include a detailed derivation of these form factors on the kagome lattice in the revised manuscript. — To obtain the smooth continuum features shown in Fig. 4 from the discrete dispersion relations, I assume that some broadening has been employed. Please elaborate.
-
On page 8, it is written that the $\pi$ “flux pattern in the triangular lattice doesn’t cause symmetry fractionalization but reverses the energy band.’’ Why?
-
In Sec. VII A.4, the authors write that “To access the spectroscopic properties of the $\mathbb{Z}_2$ liquid phase in the polar molecule systems, one can use the two-photon Raman spectroscopy [93] to obtain equivalent outcomes as the inelastic neutron scatterings for the spins.’’ This is a sweeping claim and has to be substantiated with details instead of deferring the responsibility to Ref. 93. In ultracold atomic systems, measurement of two-time correlation functions, as required for dynamical structure factors, is challenging because projective measurement collapses the wavefunction. Hence, it would be very useful if the authors could actually provide the schematic details for extracting such spectroscopic properties.
-
Finally, in discussing the fate of RE$_3$BWO$_9$ and and Ca$_3$RE$_3$(BO$_3$) under the effect of kagome lattice distortions, the authors mention the possibility that “the frustrated and long-range 3D interactions may stabilize the $\mathbb{Z}_2$ topological order in 3D. ’’ Is there any known example of a model in which this scenario occurs?
Recommendation
Ask for minor revision

---

## Round 2 · Referee Report · Anonymous (Referee 1) · 2025-6-24

Report

The authors have addressed all my concerns. I thus recommend publication in SciPost Physics.

Recommendation

Publish (meets expectations and criteria for this Journal)

---

## Round 2 · Referee Report · Anonymous (Referee 2) · 2025-7-7

Report

The authors have satisfactorily addressed all my questions and I am happy to recommend the publication of this manuscript in SciPost Physics.

Recommendation

Publish (meets expectations and criteria for this Journal)

---

## Round 2 · List of Changes

Summary of Changes
1. At the end of Sec. V B, a discussion about possible competing phases is added.
2. In the beginning of the 4th paragraph of Sec. I, “Since the Z2 topological order has not yet been fully discovered in any existing experiment” is revised to “While Z2 topological order has appeared in Rydberg atom experiments and quantum simulators, its evidence in real condensed matter materials has not yet been fully discovered in any existing experiment. Therefore, it is more important for us to address the key experimental features that distinguish it from other states.”
3. At the end of Sec. VII A 1, a discussion about measuring dynamical signatures of fractional excitations is added.
4. “kagomé” is replaced by “kagome”.
5. At the end of Sec. IV A, “For mz = msat/3$, recent numerical simulations suggest that the Z2 liquid phase also lives in the finite μ region, and a hidden order is reported near and above μ=0” is added.
6. Several typos are fixed.

---

## Editorial Decision

in_voting